# Direct Advantage Estimation

**Hsiao-Ru Pan**[1]* **Nico Gürtler**[1] **Alexander Neitz**[2]† **Bernhard Schölkopf**[1]
[1]Max Planck Institute for Intelligent Systems, Tübingen
[2]DeepMind

## Abstract

The predominant approach in reinforcement learning is to assign credit to actions based on the expected return. However, we show that the return may depend on the policy in a way which could lead to excessive variance in value estimation and slow down learning. Instead, we show that the advantage function can be interpreted as causal effects and shares similar properties with causal representations. Based on this insight, we propose Direct Advantage Estimation (DAE), a novel method that can model the advantage function and estimate it directly from on-policy data while simultaneously minimizing the variance of the return without requiring the (action-)value function. We also relate our method to Temporal Difference methods by showing how value functions can be seamlessly integrated into DAE. The proposed method is easy to implement and can be readily adapted by modern actor-critic methods. We evaluate DAE empirically on three discrete control domains and show that it can outperform generalized advantage estimation (GAE), a strong baseline for advantage estimation, on a majority of the environments when applied to policy optimization.

## 1 Introduction

Reinforcement learning (RL) methods aim to maximize cumulative rewards in sequential decision making problems [Sutton and Barto, 2018]. Through interactions with the environment, agents learn to identify which actions lead to the highest return. One major difficulty of this problem is that, typically, a vast amount of decision makings are involved in determining the rewards, making it complex to identify which decisions are crucial to the outcomes. This is also known as the credit assignment problem [Minsky, 1961]. A straightforward approach is to assign credit according to the expected return, which is aligned with the RL objective[Sutton and Barto, 2018]. However, the expected return accumulates all the future rewards and does not directly reflect the effect of an action. Consider, for example, the environment illustrated in Figure 1. Since the transitions at each state are independent of the actions chosen, the *effect* of any action should be immediate. The expected return conditioned on any action, however, depends on the policy of all subsequent states. This property could be undesirable because the expected return would vary significantly whenever the policy changes during training, making it a difficult target to learn.

A natural question to ask is, then, how should we define the *effect* of an action? In this work, we take inspiration from the causality literature, where the notion of causal effects is defined. We show that the advantage function can be interpreted as the causal effect of an action on the expected return. Additionally, we show that it shares similar properties with causal representations, which are expected to improve generalization and transfer learning [Schölkopf et al., 2021].

We prove that the advantage function minimizes the variance of the return under a constraint, and propose **Direct Advantage Estimation** (DAE), a method that can model and estimate the advantage

---

*Correspondence to: hpan@tuebingen.mpg.de
†Work done while at MPI-IS.

36th Conference on Neural Information Processing Systems (NeurIPS 2022).

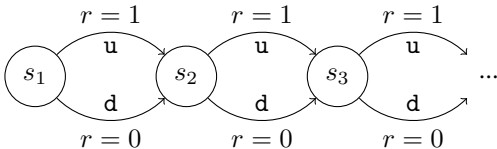

Figure 1: An example environment where the effect of an action on the return is only immediate. The agent starts in state $s_1$ and, at each step $t$, chooses either the action u (up) or d (down) with immediate rewards of 1 or 0, respectively, before transitioning into the next state $s_{t+1}$.

function directly from sampled trajectories unlike previous methods that rely on learning either the value function or the $Q$-function beforehand. Furthermore, we show that DAE can be combined with a value function approximator and be updated in a way similar to Temporal Difference methods [Sutton, 1988]. We then illustrate how DAE can be integrated into modern deep actor-critic methods using the example of Proximal Policy Optimization (PPO) [Schulman et al., 2017]. Finally, we test our method empirically on three discrete domains including (1) a synthetic environment, (2) the MinAtar suite [Young and Tian, 2019] and (3) the Arcade Learning Environment [Bellemare et al., 2013], and demonstrate that DAE outperforms Generalized Advantage Estimation (GAE) [Schulman et al., 2015b] on most of them.

The main contributions of this paper are

- We analyze the properties of the advantage function and show how they are linked to causality.

- We develop DAE, a novel on-policy method for estimating the advantage function directly, and compare it empirically with GAE using PPO.

## 2 Background

We consider a discounted Markov Decision Process $(\mathcal{S}, \mathcal{A}, P, r, \gamma)$ with finite state space $\mathcal{S}$, finite action space $\mathcal{A}$, transition probability $P(s'|s, a)$, expected reward function $r : \mathcal{S} \times \mathcal{A} \to \mathbb{R}$, and discount factor $\gamma \in [0, 1)$. A policy is a function $\pi : \mathcal{S} \times \mathcal{A} \to \mathbb{R}$ with $\pi(a|s)$ denoting the probability of action $a$ given state $s$. A trajectory $\tau = (s_0, a_0, s_1, a_1, ...)$ is said to be sampled from a policy $\pi$ if $a_t \sim \pi(\cdot|s_t)$ and $s_{t+1} \sim P(\cdot|s_t, a_t)$. We denote the return of a trajectory by $G(\tau) = \sum_{t \geq 0} \gamma^t r_t$, where $r_t = r(s_t, a_t)$. The goal of an RL agent is to find a policy $\pi^*$ that maximizes the expected return, i.e., $\pi^* = \arg\max_\pi \mathbb{E}_{\tau \sim \pi} [G(\tau)]$. Given a policy $\pi$, we define the value function by $V^\pi(s) = \mathbb{E}_{\tau \sim \pi} [G(\tau)|s_0=s]$, the action-value function by $Q^\pi(s, a) = \mathbb{E}_{\tau \sim \pi} [G(\tau)|s_0=s, a_0=a]$, and the advantage function by $A^\pi(s, a) = Q^\pi(s, a) - V^\pi(s)$ [Sutton and Barto, 2018, Baird, 1993]. For simplicity, we shall omit the superscript. These functions quantify how good a certain state (state-action pair) is, which can be used to improve the policy. In practice, however, they are usually unknown to the agent, and have to be learned from its interactions with the environment. Below, we briefly summarize how they are typically learned.

**Monte Carlo (MC) method.** By definition, an unbiased way to estimate $V(s)$ is by sampling trajectories starting from state $s$ and averaging the sample returns $V(s) \approx \frac{1}{n} \sum_{i=1}^n G(\tau_i)$. While being conceptually simple and unbiased, MC methods are known to suffer from high variance, which can cause slow learning.

**Temporal Difference (TD) learning.** A different approach based on bootstrapping is proposed by Sutton [1988]. In this approach, we begin with an initial guess of the value function $V$ and improve our estimates iteratively by sampling transitions $(s, a, r, s')$ and updating our current estimates to match the bootstrapping targets via $V(s) \leftarrow V(s) + \alpha(r + V(s') - V(s))$. This is also known as the TD(0) method. Unlike MC methods, TD methods have lower variance at the cost of being biased due to their use of bootstrapping targets. Methods based on bootstrapping have been shown to converge faster than MC methods empirically and have been widely adapted in recent deep RL methods [Mnih et al., 2015, 2016, Schulman et al., 2017].

One disadvantage of the TD(0) method is that it only propagates the reward one step at a time, which can be slow when the rewards are sparse. Instead, one can utilize the rewards that are up to $n$-steps ahead and update value estimates by $r_t + \gamma r_{t+1} + ... + \gamma^n V(s_{t+n})$. This allows the reward information to propagate faster to relevant state-action pairs. Additionally, this view unifies both methods in the sense that TD(0) methods correspond to the case when $n = 1$ while $n \to \infty$ recovers MC methods. Choosing the parameter $n$ can be seen as a bias and variance trade-off, where lower $n$ relies more on bootstrapping which could be biased while higher $n$ relies more on the sampled trajectory which is less biased but could lead to larger variance. In TD($\lambda$), these $n$-step estimations are further combined to produce a more robust estimate by averaging them exponentially with $\lambda^n$ [Sutton, 1988, Watkins, 1989].

**TD errors and the advantage function.**  One way to estimate $A(s, a)$ is to sample a transition $(s, a, r, s')$ and compute $A(s, a) = Q(s, a) - V(s) \approx r + \gamma V(s') - V(s)$. The term $r + \gamma V(s') - V(s)$ is known as TD Error and was introduced in the TD Learning algorithm to update the value estimates. This method can also be combined with TD($\lambda$) to produce more accurate estimates of the advantage function [Schulman et al., 2015b].

**Policy optimization.**  One important application of the advantage function is in policy optimization. Williams [1992], Sutton et al. [2000] showed that, for a parameterized policy $\pi_\theta$, the gradient of the RL objective is given by

$$\nabla_\theta \mathbb{E}_{\tau \sim \pi_\theta} [G(\tau)] = \sum_{s \in \mathcal{S}} d^{\pi_\theta}(s) \sum_{a \in \mathcal{A}} (Q(s, a) - b(s)) \nabla_\theta \pi_\theta(a|s), \tag{1}$$

where $d^\pi(s) = \sum_{t \geq 0} \gamma^t p(s_t = s)$ and $b(s)$ is an arbitrary function of the state known as the baseline function. The baseline function can be seen as a control variate, which can be used to reduce the variance of the gradient estimator when chosen carefully [Greensmith et al., 2004]. One common choice of $b(s)$ is the value function $V$, which results in the advantage function.

In a different approach, Kakade and Langford [2002] showed that the difference between the expected return of two policies is directly related to the advantage function, which paved the way for some of the most popular policy optimization methods [Schulman et al., 2015a, 2017].

## 3   Revisiting the advantage function

We begin by examining the motivating example in Figure 1. In this example, it is clear that the optimal decision depends solely on the immediate reward at each state, since the transitions are independent of the actions. However, if we calculate the $Q$-function, we get

$$Q^\pi(s_t, a_t) = \begin{cases} 1 + \sum_{t' > t} \gamma^{t'-t} \pi(\mathtt{u}|s_{t'}), \ a_t = \mathtt{u} \\ 0 + \sum_{t' > t} \gamma^{t'-t} \pi(\mathtt{u}|s_{t'}), \ a_t = \mathtt{d} \end{cases} \tag{2}$$

which shows that the $Q$-values at each state depend on the policy evaluated at all subsequent states despite the actions having only immediate effects on the expected return. This property can be undesirable as the policy is typically being optimized and varied, which could lead to variations in the $Q$-values and make previous estimates unreliable. This is also known as the problem of *distribution shift*, where the data generating distribution differs from the distribution we are making predictions on. Similar problems have also been studied in offline RL, where distribution shift can cause $Q$-function estimates to be unreliable Levine et al. [2020], Kumar et al. [2020], Fujimoto et al. [2019]. One source of this problem is that the expected return by nature associates an action with all its subsequent rewards, whether they are causally related or not. Alternatively, we would like to focus on the *effect* of an action. But how should we define it?

Questions like this are central to the field of causality and have been studied extensively [Holland, 1986, Pearl, 2009]. We draw our inspiration from Splawa-Neyman et al. [1990] and Rubin [1974] in which the authors developed a framework, now known as the Neyman-Rubin model, to estimate causal effects of treatments. It defines the *causal effect* of a treatment to be the difference between (1) the outcome of a trial resulting from one treatment, and (2) what would have happened if we had chosen another treatment. The work focused mainly on the problem of counterfactuals, which we shall not delve into. Instead, we focus simply on its definition of causal effects. In RL, we concern

ourselves with the effect of each action on the expected return. However, the action space usually consists of more than two actions, so it is ambiguous as to which action we should compare to. One way to resolve this is to compare each action with what would have happened normally [Halpern and Hitchcock, 2015]. This way, we can define the causal effect of an action $a$ on some quantity $X$ at state $s$ by

$$\mathbb{E}\left[X|s,a\right] - \mathbb{E}\left[X|s\right]. \tag{3}$$

If we choose $X$ to be the return, then

$$\mathbb{E}\left[G(\tau)|s_t = s, a_t = a\right] - \mathbb{E}\left[G(\tau)|s_t = s\right] = Q(s,a) - V(s) = A(s,a) \tag{4}$$

is precisely the advantage function, suggesting that it can be interpreted as the causal effect on the expected return. Readers with more exposure to the causality literature may wonder why machinery such as intervention or counterfactual are not required to define such quantities. This is possible mainly because we focus on fully observable environments where confounders are not present, and we expect more sophisticated treatments would be required if one wishes to extend similar ideas to more general problems (e.g., partially observable environments [Bareinboim et al., 2015, Kumor et al., 2021]).

If we calculate the advantage function for the example environment in Figure 1, we get

$$A^\pi(s,a) = \begin{cases} 1 - \pi(\mathtt{u}|s), & a = \mathtt{u} \\ 0 - \pi(\mathtt{u}|s), & a = \mathtt{d} \end{cases} \tag{5}$$

which now depends solely on the policy evaluated at the state $s$. This is because, in this example, the future state distribution is independent of the actions, so the conditional expectation cancels out, leaving only the immediate term. More generally, we only need the expected return for the future state to be independent of the current action.

**Proposition 1.** Given $t' > t$ such that $\mathbb{E}\left[V(s_{t'})|s_t, a_t\right] = \mathbb{E}\left[V(s_{t'})|s_t\right]$, we have

$$A(s_t, a_t) = \mathbb{E}\left[\sum_{k=t}^{t'-1}\gamma^{k-t}r_k \;\middle|\; s_t, a_t\right] - \mathbb{E}\left[\sum_{k=t}^{t'-1}\gamma^{k-t}r_k \;\middle|\; s_t\right] \tag{6}$$

See Appendix A for proofs. This proposition shows that the advantage function is *local*, in the sense that the advantage of an action is only dependent on the policy of subsequent states to the point where the action stops being relevant to future rewards. This property is strikingly similar to the Sparse Mechanism Shift hypothesis [Schölkopf et al., 2021], which states that small distribution changes should only lead to sparse or local changes if the representation is causal/disentangled.

The condition of Proposition 1 may be too strict to hold in most cases. As a relaxation, we hypothesize that, for a large class of problems of interest, the advantage function is more stable under policy variations compared to the $Q$-function, making it an easier target to learn. This may partially explain performance improvements reported by Baird [1993], Wang et al. [2016b]. In Figure 2, we verify this empirically using the Breakout environment from the MinAtar suite [Young and Tian, 2019] by tracking

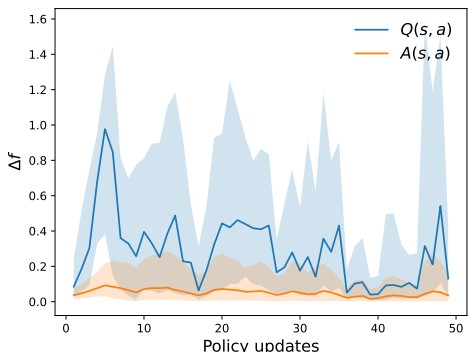

Figure 2: $\Delta f = |f^{\pi_i} - f^{\pi_{i-1}}|$ estimated using MC simulations between each policy update computed with 3072 state-action pairs sampled throughout training. Lines and shadings represent the median and the interquartile range over all sampled state-action pairs. See Appendix C for details.

the variations of $Q$ and $A$ throughout the training of a PPO agent. We can see that the advantage function remains more stable over the course of training compared to the $Q$-function. From a more theoretical point of view, this softer condition is satisfied when $p(s_{t'}|s_t, a_t) \approx p(s_{t'}|s_t)$ for some $t' > t$. This can happen when, for example, the Markov chain induced by the policy approaches

a stationary distribution; as another example, if the environment contains certain bottleneck states (e.g., doorways), that are traversed with high probability, then the state distribution after passing those states would depend weakly on the actions before passing them. On the other hand, if the environment is "tree-like", where actions lead to completely different branches, then the advantage function would also suffer from instability.

The question we are now interested in is how to learn the advantage function. In previous work, estimating the advantage function typically relied on using TD errors. This method also faces the problem of relying on an estimate of the value function that suffers from strong policy dependency. Instead, we seek methods which could directly model the advantage function and estimate it from data. Since both $Q$ and $V$ can be learned with bootstrapping via TD learning, one might wonder if there is an equivalent for the advantage function. Here we give a counterexample showing that the advantage function cannot be learned in the same way.

Consider the environment in Figure 3. In this example, the undiscounted advantage function for $s$ is also equal to Equation 5, while the advantages for both $s_u$ and $s_d$ are zero because there is only a single action in their action spaces. Since the immediate rewards and the advantages for the next states are all zero for the state $s$, we conclude that it is not possible to learn the advantages of $s$ based solely on the immediate reward and the advantages of the next state as in TD learning. An intuitive explanation for this failure is that, unlike the value function, which contains information about all future rewards, the advantage function only models the causal effect of each action; therefore, if the next action is not causal to a distant reward, then the information about that reward is lost.

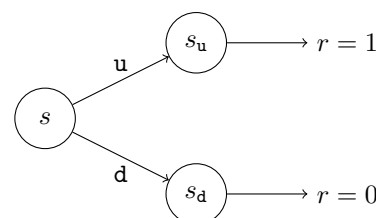

Figure 3: A synthetic environment demonstrating the impossibility of learning $A$ from bootstrapping. An agent begins in state $s$ with actions u and d which transition the agent into states $s_u$ and $s_d$, respectively, with no immediate reward. In both $s_u$ and $s_d$, there is only one action with immediate rewards 1 and 0, respectively.

## 4  Direct Advantage Estimation

In this section, we present the main contribution of this paper. We develop a simple method to model and estimate the advantage function directly from sampled trajectories.

Before we begin, we first observe that the advantage function satisfies the equality $\sum_{a \in \mathcal{A}} \pi(a|s) A^\pi(s, a) = 0$. Functions with this property, as it turns out, are useful for the development of our method, so we provide a formal definition.

**Definition 1.** Given a policy $\pi$, we say a function $f : \mathcal{S} \times \mathcal{A} \to \mathbb{R}$ is $\pi$-centered if for all states $s \in \mathcal{S}$

$$\sum_{a \in \mathcal{A}} \pi(a|s) f(s, a) = 0. \tag{7}$$

As discussed in Section 2, MC and $n$-step methods often suffer from excessive variance due to their sampling nature. It is therefore desirable to find unbiased low-variance estimates of the return. One interesting property of $\pi$-centered functions is that, if we transform the reward by $r_t' = r_t - \hat{A}(s_t, a_t)$ with any $\pi$-centered function $\hat{A}$, then the expected return remains unchanged, that is $\mathbb{E}[G(\tau)] = \mathbb{E}[G'(\tau)]$, where $G'(\tau) = \sum_{t \geq 0} \gamma^t r_t'$. This can be seen as a form of reward shaping [Ng et al., 1999], where the expected return is now invariant to the choice of $\hat{A}$ for a fixed policy. Now, consider the variance of the transformed return $G'(\tau)$: we have $\mathbf{Var}[G'(\tau)] = \mathbb{E}[G'(\tau)^2] - \mathbb{E}[G(\tau)]^2$. In other words, minimizing its variance with respect to $\hat{A}$ is equivalent to minimizing the following quantity,

$$\mathbb{E}\left[\left(\sum_{t=0}^{\infty} \gamma^t \left(r_t - \hat{A}(s_t, a_t)\right)\right)^2\right]. \tag{8}$$

Minimizing the variance of the return this way has the additional benefit of simultaneously obtaining the advantage function without having to learn the (action-)value function first, as we show in our main theorem.

**Theorem 1.** Given a policy $\pi$ and time step $t \geq 0$, we say $(s, a) \in \mathcal{S} \times \mathcal{A}$ or $s \in \mathcal{S}$ is reachable within $t$ if there exists $0 \leq t' \leq t$ such that $p(s_{t'}=s, a_{t'}=a) > 0$ or $p(s_{t'}=s) > 0$, respectively. Let $F_\pi$ be the set of all $\pi$-centered functions, $\hat{A}_t = \hat{A}(s_t, a_t)$, and

$$\hat{A}^* = \arg\min_{\hat{A} \in F_\pi} \mathbb{E}\left[\left(G(\tau) - \sum_{t'=0}^{t} \gamma^{t'} \hat{A}_{t'}\right)^2\right],$$

then for all $(s, a)$ that are reachable within $t$, we have $\hat{A}^*(s, a) = A^\pi(s, a)$.

See Appendix A for proofs. Note that the first term inside the expectation is equal to the return, while the summation in the second term is only up to step $t$. We observe two interesting properties of the advantage function from this theorem. (1) *Disentanglement*: If $(s, a)$ is reachable within $t$, then $\hat{A}^*(s, a) = A^\pi(s, a)$ remains invariant when we apply the theorem to any $\hat{t} > t$, suggesting that additional $\hat{A}_{t'}$'s do not interfere with existing ones. (2) *Additivity*: One intuitive way to understand the term $\sum_{t'} \gamma^{t'} \hat{A}_{t'}$ is to view it as the total effect from the sequence of actions, which is simply the sum of individual effects $\hat{A}_{t'}$. A similar property was also pointed out by Kakade and Langford [2002], where they showed how the consecutive sum of the advantage function is related to the expected return.

This theorem also suggests a natural way to estimate the advantage function directly from sampled trajectories $\tau_1, \tau_2, \ldots, \tau_N$ by minimizing the following constrained loss function

$$L(\theta) = \frac{1}{N} \sum_{\tau=\tau_1,\ldots,\tau_N} \left(\sum_{t=0}^{\infty} \gamma^t \left(r_t - \hat{A}_\theta\left(s_t, a_t\right)\right)\right)^2 \quad \text{s.t.} \sum_a \pi(a|s)\hat{A}_\theta(s, a) = 0, \quad (9)$$

In practice, the constraint can be forced upon a general function approximator $f_\theta$, such as a neural network, by choosing $\hat{A}_\theta(s, a) = f_\theta(s, a) - \sum_{a \in \mathcal{A}} \pi(a|s) f_\theta(s, a)$. In our method, neither $V$ nor $Q$ is required a priori. Instead, the advantage function is directly modeled and estimated from data, and hence we name our method **Direct Advantage Estimation** (DAE). Unlike learning $V$ or $Q$ where one aims to find a mapping between states (state-action pairs) and its expected return, DAE tries to regress the *sum* of $A(s, a)$ along the trajectories towards the associated returns. This can be interpreted as learning to *distribute* the return rather than to *predict* it.

One limitation of this method, similar to the MC methods, is that we have to sample trajectories until they terminate. To overcome this, we show that the value function can be seamlessly integrated into our method, which enables us to perform updates similarly to the $n$-step bootstrapping method.

**Theorem 2.** Following the notation in Theorem 1, let $V_{\text{target}}(s)$ be the bootstrapping target and $\hat{V}(s)$ be the value function to be learned, and

$$L(\hat{A}, \hat{V}) = \mathbb{E}\left[\left(\sum_{t=0}^{n-1} \gamma^t(r_t - \hat{A}_t) + \gamma^n V_{\text{target}}(s_n) - \hat{V}(s_0)\right)^2\right]. \quad (10)$$

If $\hat{V}^* = \arg\min_{\hat{V}} L(\hat{A}, \hat{V})$ and $p(s_0=s) > 0$, then for any $\hat{A} \in F_\pi$, we have

$$\hat{V}^*(s) = \mathbb{E}\left[\sum_{t=0}^{n-1} \gamma^t r_t + \gamma^n V_{\text{target}}(s_n) \,\middle|\, s_0=s\right]. \quad (11)$$

Additionally, let $w_t(s) = \gamma^{2t} p(s_t=s)$ and $W_n(s) = \sum_{t=0}^{n} w_t(s)$. If $\hat{A}^* = \arg\min_{\hat{A} \in F_\pi} L(\hat{A}, \hat{V})$ and $(s, a)$ is reachable within $t = 0$ to $t = n - 1$, then for any $\hat{V} : \mathcal{S} \to \mathbb{R}$, we have

$$\hat{A}^*(s, a) = \sum_{t=0}^{n-1} \frac{w_t(s)}{W_{n-1}(s)} \left(\mathbb{E}\left[r_t + \cdots + \gamma^{n-t-1} r_{n-1} + \gamma^{n-t} V_{\text{target}}(s_n) \,\middle|\, s_t=s, a_t=a\right]\right.$$

$$\left. - \mathbb{E}\left[r_t + \cdots + \gamma^{n-t-1} r_{n-1} + \gamma^{n-t} V_{\text{target}}(s_n) \,\middle|\, s_t=s\right]\right) \quad (12)$$

Note that in Equation 12, the minimizer $\hat{A}^*$ depends only on the bootstrapping target $V_{\text{target}}$ but not the estimated value function $\hat{V}$, and vice versa. One might notice that Equation 11 is actually the multi-step update for the value function. This suggests that we can update $\hat{V}$ iteratively by letting $V_{\text{target}} = \hat{V}_{k-1}$ and $\hat{V}_k(s) = \hat{V}^*(s) = \mathbb{E}\left[\sum_{t=0}^{n-1}\gamma^t r_t + \gamma^n \hat{V}_{k-1}(s_n)\Big| s_0{=}s\right]$, which replicates multi-step TD learning. Similarly, Equation 12 can be understood as a multi-step estimation for the advantage function with bootstrapping. At first sight, this may seem to suffer from the same problem of spurious policy dependence due to the use of value functions (see discussion in Section 3). We argue, however, that this can be mitigated by using long backup horizons (i.e. $n \gg 1$) and the assumption that actions have local effects (i.e. $p(s_{t'}|s_t, a_t) \approx p(s_{t'}|s_t)$ for some $t' > t$). If the backup horizon is large enough such that $p(s_{t+n}|s_t, a_t) \approx p(s_{t+n}|s_t))$, then Equation 12 suggests that the dependence on $V_{\text{target}}$ can be reduced since $\mathbb{E}[V_{\text{target}}(s_{t+n})|s_t, a_t] - \mathbb{E}[V_{\text{target}}(s_{t+n})|s_t] \approx 0$.

In practice, we can sample $n$-step trajectories and use Equation 10 as the loss function to estimate both the advantage function and the value function simultaneously. However, naively applying the theorem this way can cause slow learning because the value function is learned only for the first state in each trajectory and distant state-action pairs are heavily discounted. To mitigate this, we further utilize each trajectory by treating each sub-trajectory as a trajectory sampled from the same policy. For example, let $(s_0, a_0, \ldots, s_n)$ be an $n$-step trajectory. Then we treat $(s_i, a_i, \ldots, s_n)$ for all $i \in \{0, \ldots, n-1\}$ also as a trajectory sampled from the same policy. This can be seen as repeated applications of Theorem 2, which leads us to the following loss function

$$L_A(\theta, \phi) = \mathbb{E}\left[\sum_{t=0}^{n-1}\left(\sum_{t'=t}^{n-1}\gamma^{t'-t}\left(r_{t'} - \hat{A}_\theta(s_{t'}, a_{t'})\right) + \gamma^{n-t}V_{\text{target}}(s_n) - \hat{V}_\phi(s_t)\right)^2\right]. \quad (13)$$

In practice we may use a single function approximator with parameters $\theta$ to model both $\hat{A}$ and $\hat{V}$ simultaneously as done by Wang et al. [2016b], which reduces the loss function to $L_A(\theta)$.

## 5 Experiments

We evaluate our method empirically by comparing the performance of policy optimization between DAE and GAE on three different sets of discrete control tasks. (1) A synthetic environment based on Figure 1 where the true advantage function and the expected return are known, (2) the MinAtar [Young and Tian, 2019] suite, a set of environments inspired by Atari games with similar dynamics but simpler observation space, and (3) the Arcade Learning Environment (ALE) [Bellemare et al., 2013], a set of tasks with diverse mechanisms and high-dimensional observation spaces.

### 5.1 Synthetic environment

We consider a finite variant of the environment shown in Figure 1 with state space $\mathcal{S} = \{s_1, ..., s_{128}\}$, and train agents using a simple actor-critic algorithm where in each iteration we (1) sample trajectories, (2) estimate the advantages of the sampled state-action pairs, and (3) perform a single policy gradient step weighted by the estimated advantages. For both methods, we use the same network architectures and hyperparameters to train the agents, see Appendix C for more details. We assess the quality of the advantage estimation through mean squared error (MSE) between the estimated and the true advantage function (Equation 5) on the sampled state-action pairs in each iteration. Additionally, we evaluate the learned policy by computing the true (undiscounted) expected return by $\mathbb{E}\left[\sum_i r_i\right] = \sum_i \pi(\text{u}|s_i)$. Note that the optimal policy in this case is $\pi(\text{u}|\cdot) = 1$ with expected return $\sum_i \pi(\text{u}|s_i) = 128$.

The results in Figure 4 demonstrate that DAE can approximate the true advantage function accurately, while GAE struggles. The inability of GAE to accurately estimate the advantage function is caused by the variance of the $n$-step returns used in the estimates, which slowly decreases as the policy converges to the deterministic greedy policy. For policy optimization, we see that GAE performs better in the early phase of training, but is later surpassed by DAE. This is likely because estimates from GAE include unbiased estimates of $n$-step returns that are useful to policy optimization, while DAE relies entirely on the approximated function, which is heavily biased at the beginning. As DAE becomes more accurate later in training, we see substantial gains in performance, which demonstrates a drawback of using high variance estimates from GAE.

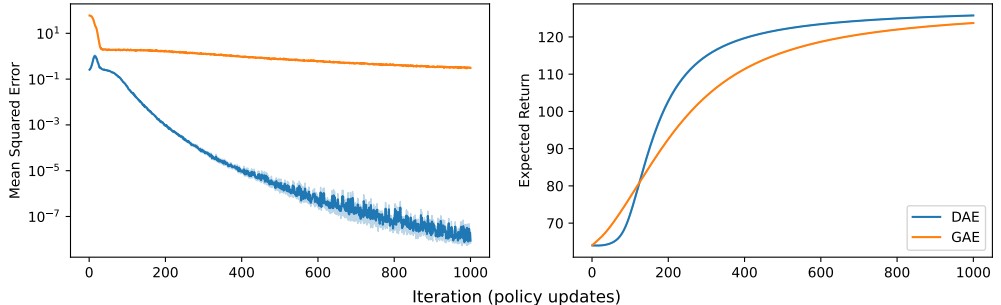

Figure 4: Results for the synthetic environment. Lines and shadings represent the mean and one standard error over 100 random seeds. Left: Mean squared error (in log scale) between the true and estimated advantage function on the state-action pairs sampled in each iteration. Right: The (undiscounted) expected return of the policy.

---

**Algorithm 1** PPO with DAE (shared network)

1: Initialize $\hat{A}$, $\hat{V}$ and the policy $\pi$ with $\theta$
2: **while** not converged **do**
3:     $D = \{\}$
4:     **for** $i = 1, \ldots, N_{\text{actors}}$ **do**
5:         Sample $n$-step trajectory $\tau_i = (s_0, a_0, r_0, \ldots, s_n)$ with policy $\pi$
6:         $D \leftarrow D \cup \{\tau_i\}$
7:     **end for**
8:     $\phi \leftarrow \texttt{copy}(\theta)$
9:     $V_{\text{target}}, \mu \leftarrow V_\phi, \pi_\phi$
10:     **for** $i = 1, \ldots, N_{\text{updates}}$ **do**
11:         Sample a mini-batch of $n$-step trajectories $B$ from $D$
12:         Compute $\hat{A}(s, a)$ for all $(s, a) \in B$ (centered according to $\mu$)
13:         Compute PPO clipping loss $L_\pi$ (Equation 49) with $\texttt{stop\_gradient}(\hat{A}(s, a))$, $\mu$, and $\pi$
14:         Compute DAE loss $L_A$ (Equation 13)with $\hat{A}$, $\hat{V}$ and $V_{\text{target}}$
15:         Optimize loss $L = (L_\pi + \beta_V L_A)$
16:     **end for**
17: **end while**

---

## 5.2 MinAtar & Arcade Learning Environment (ALE)

We now test DAE's performance in the more challenging domains of the MinAtar and the ALE environments using PPO as the base algorithm. We use the PPO implementation and the tuned hyperparameters from Raffin et al. [2021], Raffin [2020]. A practical implementation of PPO with DAE can be found in Algorithm 1 (see Appendix B for more details)[3]. For DAE, we tune two of the hyperparameters, namely the scaling coefficient of the value function loss and the number of epochs in each PPO iteration using the MinAtar environments, which are then fixed for the ALE experiments. Note that DAE has one less hyperparameter ($\lambda$) to tune because it does not rely on TD($\lambda$). Additionally, we increase the number of parallel actors for both methods as it substantially speeds up PPO training in terms of wall-clock time and has become the predominant way to train PPO agents in complex environments [Rudin et al., 2022, Freeman et al., 2021]. We also test how network capacities impact the performance of both methods by increasing the width of each layer in the baseline network (denoted "Wide"). For ALE, we further test a deep residual network architecture (denoted "Deep") based on Espeholt et al. [2018]. See Appendix C for a more detailed description of the network architectures and hyperparameters.

Each agent is trained independently for 10 million and 40 million in-game frames for the MinAtar and the ALE environments, respectively. We compare both methods on two metrics, (1) *Overall*: average undiscounted score of all training episodes, and (2) *Last*: average undiscounted score of the

---

[3]Code is available at $\texttt{https://github.com/hrpan/dae}$.

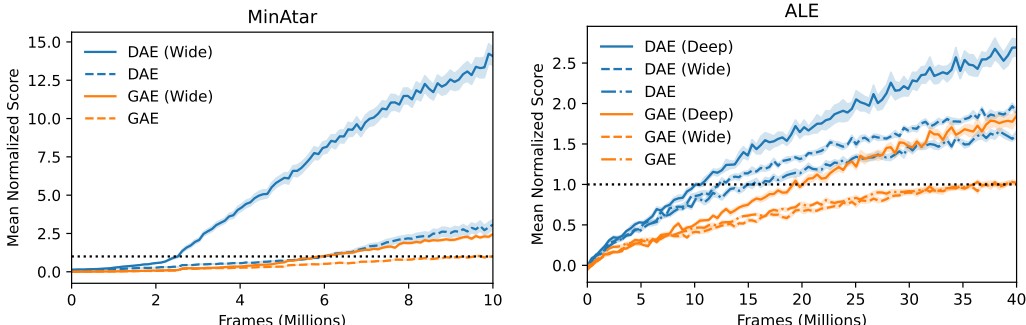

Figure 5: Mean GAE-normalized score for the MinAtar (left) and the ALE (right) experiments. The lines represent the average over all runs (50 and 10 random seeds for the MinAtar and the ALE, respectively) with each run corresponding to the average over all environments with one random seed, and the shadings correspond to one standard error of the mean. The scores are normalized based on GAE's (baseline network architecture) average score in the first 1% frames and the last 1% frames.

last 100 training episodes, as in Schulman et al. [2017]. The metrics reflect how fast an agent learns and the final performance of the learned policy, respectively.

From Table 1, we observe that, given the same network architecture, DAE outperforms GAE in all the MinAtar environments and most of the ALE environments on both metrics. In Figure 5, we present the GAE-normalized learning curves for each domain and architecture, where we also see a similar trend that DAE is much more sample efficient while achieving better final performance.[4] Results for individual environments can be found in Appendix E. Finally, we found that increasing the network capacities consistently improves DAE's performance while having less impact on GAE. This is likely because the advantage function is directly represented in DAE using the network, such that network capacities should have a larger impact on the performance. Interestingly, we see significant improvements in the MinAtar experiments by switching to the Wide network for DAE but not GAE. Note that this gap cannot be explained by the increased representation power in the policy network alone, or we should also see significant improvements with GAE. Therefore, it suggests that the improvements are likely originated from the ability to more accurately approximate the advantage function. In the ALE experiments, we also observe such jumps in performance for some environments (see Appendix C), but for most environments, both DAE and GAE improved significantly. This suggests that the improved performance can be attributed to the increased capacities in the policy networks, which benefit both methods.

# 6 Related work

The advantage function was first introduced by Baird [1993] to tackle problems with fine-grained time steps, where $Q$-Learning could be slow. In policy optimization methods, the advantage function has also been widely used as an alternative to $Q$-functions to lower the variance of policy updates [Greensmith et al., 2004, Kakade and Langford, 2002, Sutton et al., 2000]. More recently, Wang et al. [2016b] developed a neural network architecture which can jointly estimate the value function and the advantage function to achieve better performance with DQN [Mnih et al., 2015]. Our work complements theirs by justifying the use of policy centered functions in estimating the advantage function, and showing how it can be combined with multi-step learning. Schulman et al. [2015b] proposed GAE, a method which combines TD($\lambda$) and TD error to estimate the advantage function, and demonstrated strong empirical results in policy optimization.

The link between causality and credit assignment in RL has also been explored in Mesnard et al. [2020], where the authors used the idea of counterfactuals to construct a future-conditional value function to reduce the variance of policy gradient updates. Connections between causal effects and the

---

[4]Hard exploration environments such as Montezuma's Revenge are excluded in the normalization because the GAE baseline does not improve consistently.

Table 1: Number of environments won by each method. The metrics were averaged across all runs before comparison. Results are counted as "Similar" if their 1 standard error interval overlaps. See Appendix E for detailed results.

| Model Size | Method | Metric | | | |
| | | Overall | | Last | |
| | | MinAtar | Atari | MinAtar | Atari |
| --- | --- | --- | --- | --- | --- |
| Baseline | GAE | 0 | 10 | 0 | 11 |
| | DAE | 5 | 32 | 5 | 30 |
| | Similar | 0 | 7 | 0 | 8 |
| Wide | GAE | 0 | 8 | 0 | 9 |
| | DAE | 5 | 35 | 5 | 34 |
| | Similar | 0 | 6 | 0 | 6 |
| Deep | GAE | - | 6 | - | 8 |
| | DAE | - | 35 | - | 32 |
| | Similar | - | 8 | - | 9 |

advantage function were also studied by Corcoll and Vicente [2020], where they model the dynamics of environments in a hierarchical manner with a differently defined advantage function.

## 7 Conclusions and future directions

In this work, we tackled the problem of credit assignment in the RL setting. Inspired by previous studies in causality, we found that the advantage function can be interpreted as the causal effect of an action on the expected return. We then proposed DAE, a novel method that can directly model the advantage function and estimate it from data. Our experiments in three different domains demonstrated that DAE can converge to the true advantage function, improve the performance of policy optimization compared to GAE in most environments, while also enjoying better scalability with increased network capacities.

Here we note some possible extensions to DAE. (1) Environments with confounders (e.g., partially observable MDPs) are common in real-world applications, and causal inference is known to be effective in such problems [Bareinboim et al., 2015, Tennenholtz et al., 2020, Kumor et al., 2021]. We expect similar ideas based on the causal effect as proposed in this work may also be useful in those settings. (2) Due to the difficulty of enforcing the centering constraint (Def. 1) in continuous domains, we currently restrict ourselves to discrete action space domains. One potential solution to this problem was outlined in Wang et al. [2016a] where they used a sampling method to approximate the centering step for continuous action spaces. (3) In this work we assumed the data is on-policy, an interesting extension would be to use importance sampling techniques such as Retrace [Munos et al., 2016] or V-Trace [Espeholt et al., 2018] to extend DAE to off-policy settings.

## Acknowledgments

The authors thank the International Max Planck Research School for Intelligent Systems (IMPRS-IS) for supporting Hsiao-Ru Pan and Alexander Neitz. This work was supported by the German Federal Ministry of Education and Research (BMBF): Tübingen AI Center, FKZ: 01IS18039B, and by the Machine Learning Cluster of Excellence, EXC number 2064/1 – Project number 390727645.

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
