# A  Proofs

Throughout this section, we use $p(s_t{=}s, a_t{=}a)$ to denote the probability of the state-action pair at time step $t$ being equal to $(s, a)$, and the probability of a trajectory by $p(\tau) = p(s_0, a_0, s_1, a_1, ...)$.

## A.1  Proof of Proposition 1

*Proof.* By definition

$$Q^\pi(s_t, a_t) = \mathbb{E}\left[\sum_{k \geq t} \gamma^{k-t} r_k \,\middle|\, s_t, a_t\right] \tag{14}$$

$$= \mathbb{E}\left[\sum_{k=t}^{t'-1} \gamma^{k-t} r_k \,\middle|\, s_t, a_t\right] + \gamma^{t'-t} \mathbb{E}\left[\sum_{k \geq t'} \gamma^{k-t'} r_k \,\middle|\, s_t, a_t\right] \tag{15}$$

$$= \mathbb{E}\left[\sum_{k=t}^{t'-1} \gamma^{k-t} r_k \,\middle|\, s_t, a_t\right] + \gamma^{t'-t} \mathbb{E}\left[V(s_{t'})|s_t, a_t\right] \tag{16}$$

Similarly, we have

$$V^\pi(s_t) = \mathbb{E}\left[\sum_{k=t}^{t'-1} \gamma^{k-t} r_k \,\middle|\, s_t\right] + \gamma^{t'-t} \mathbb{E}\left[V(s_{t'})|s_t\right] \tag{17}$$

If $\mathbb{E}\left[V(s_{t'})|s_t, a_t\right] = \mathbb{E}\left[V(s_{t'})|s_t\right]$, then

$$A^\pi(s_t, a_t) = Q^\pi(s_t, a_t) - V^\pi(s_t) \tag{18}$$

$$= \mathbb{E}\left[\sum_{k=t}^{t'-1} \gamma^{k-t} r_k \,\middle|\, s_t, a_t\right] - \mathbb{E}\left[\sum_{k=t}^{t'-1} \gamma^{k-t} r_k \,\middle|\, s_t\right] \tag{19}$$

$\square$

## A.2  Proof of Theorem 1

*Proof.* For simplicity, we denote $\hat{A}$ by $f$. Notice that the condition $f \in F_\pi$ is equivalent to the equality constraint

$$\sum_{a \in \mathcal{A}} \pi(a|s) f(s, a) = 0 \quad \forall s \in \mathcal{S}. \tag{20}$$

Using the method of Lagrange multipliers, we have the Lagrangian

$$\mathcal{L} = \mathbb{E}_{\tau \sim \pi}\left[\left(G(\tau) - \sum_{t'=0}^{t} \gamma^{t'} f_{t'}\right)^2\right] + \sum_{s \in \mathcal{S}} \lambda_s \sum_{a \in \mathcal{A}} \pi(a|s) f(s, a) \tag{21}$$

with the shorthand $f_{t'} = f(s_{t'}, a_{t'})$. If we differentiate it with respect to $f(s', a')$, then

$$\frac{\partial \mathcal{L}}{\partial f(s', a')} = -2 \mathbb{E}_{\tau \sim \pi}\left[\left(G(\tau) - \sum_{t'=0}^{t} \gamma^{t'} f_{t'}\right)\left(\sum_{k=0}^{t} \gamma^k \frac{\partial f_k}{\partial f(s', a')}\right)\right] + \lambda_{s'} \pi(a'|s') \tag{22}$$

$$= -2 \sum_{k=0}^{t} \gamma^k \mathbb{E}_{\tau \sim \pi}\left[\left(G(\tau) - \sum_{t'=0}^{t} \gamma^{t'} f_{t'}\right) \frac{\partial f_k}{\partial f(s', a')}\right] + \lambda_{s'} \pi(a'|s') \tag{23}$$

Note that

$$\frac{\partial f_k}{\partial f(s', a')} = \mathbb{I}(s_k = s, a_k = a) = \begin{cases} 1, & \text{if } (s_k, a_k) = (s', a') \\ 0, & \text{otherwise} \end{cases} \tag{24}$$

Hence,

$$\mathop{\mathbb{E}}_{\tau\sim\pi}\left[\left(G(\tau)-\sum_{t'=0}^{t}\gamma^{t'}f_{t'}\right)\frac{\partial f_k}{\partial f(s',a')}\right] \tag{25}$$

$$=\mathop{\mathbb{E}}_{\tau\sim\pi}\left[\left(G(\tau)-\sum_{t'=0}^{t}\gamma^{t'}f_{t'}\right)\mathbb{I}(s_k=s,a_k=a)\right] \tag{26}$$

$$=p(s_k=s',a_k=a')\mathop{\mathbb{E}}_{\tau\sim\pi}\left[G(\tau)-\sum_{t'=0}^{t}\gamma^{t'}f_{t'}\,\middle|\,s_k=s',a_k=a'\right] \tag{27}$$

If we sum over all $a' \in \mathcal{A}$, since $p(s_k=s', a_k=s') = p(s_k=s')\pi(a'|s')$, then

$$\sum_{a'\in\mathcal{A}}\frac{\partial\mathcal{L}}{\partial f(s',a')}=-2\sum_{k=0}^{t}\gamma^k p(s_k=s')\mathop{\mathbb{E}}_{\tau\sim\pi}\left[G(\tau)-\sum_{t'=0}^{t}\gamma^{t'}f_{t'}\,\middle|\,s_k=s'\right]+\lambda_{s'}=0 \tag{28}$$

which implies

$$\lambda_{s'}=2\sum_{k=0}^{t}\gamma^k p(s_k=s')\mathop{\mathbb{E}}_{\tau\sim\pi}\left[G(\tau)-\sum_{t'=0}^{t}\gamma^{t'}f_{t'}\,\middle|\,s_k=s'\right]. \tag{29}$$

Substituting this back in Equation 22, we get, assuming $\pi(a'|s') > 0$,

$$\sum_{k=0}^{t}\gamma^k p(s_k=s')\left(\mathop{\mathbb{E}}_{\tau\sim\pi}\left[G(\tau)-\sum_{t'=0}^{t}\gamma^{t'}f_{t'}\,\middle|\,s_k=s',a_k=a'\right]-\mathop{\mathbb{E}}_{\tau\sim\pi}\left[G(\tau)-\sum_{t'=0}^{t}\gamma^{t'}f_{t'}\,\middle|\,s_k=s'\right]\right)=0 \tag{30}$$

From the Markov property, we have

$$\mathop{\mathbb{E}}_{\tau\sim\pi}\left[G(\tau)-\sum_{t'=0}^{t}\gamma^{t'}f_{t'}\,\middle|\,s_k=s',a_k=a'\right]-\mathop{\mathbb{E}}_{\tau\sim\pi}\left[G(\tau)-\sum_{t'=0}^{t}\gamma^{t'}f_{t'}\,\middle|\,s_k=s'\right] \tag{31}$$

$$=\gamma^k\left(\mathop{\mathbb{E}}_{\tau\sim\pi}\left[G(\tau)-\sum_{t'=0}^{t-k}\gamma^{t'}f_{t'}\,\middle|\,s_0=s',a_0=a'\right]-\mathop{\mathbb{E}}_{\tau\sim\pi}\left[G(\tau)-\sum_{t'=0}^{t-k}\gamma^{t'}f_{t'}\,\middle|\,s_0=s'\right]\right) \tag{32}$$

$$=\gamma^k\left(Q(s',a')-f(s',a')-V(s')\right) \tag{33}$$

If $s'$ is reachable within $t$ (i.e., $p(s_k = s') > 0$ for some $k$), then

$$f(s',a')=Q(s',a')-V(s') \tag{34}$$

$\square$

### A.3   Proof of Theorem 2

*Proof.* We denote $\hat{A}$ by $f$ and $V_{\text{target}}$ by $U$, and use the method of Lagrange multipliers. Consider the Lagrangian

$$\mathcal{L}=\mathop{\mathbb{E}}_{\tau\sim\pi}\left[\left(\sum_{t'=0}^{t-1}\gamma^{t'}r'_{t'}+\gamma^t U(s_t)-\hat{V}(s_0)\right)^2\right]+\sum_{s\in\mathcal{S}}\lambda_s\sum_{a\in\mathcal{A}}\pi(a|s)f(s,a) \tag{35}$$

Let's first consider the minimum for $\hat{V}$,

$$\frac{\partial\mathcal{L}}{\partial\hat{V}(s')}=-2p(s_0=s')\sum_{\tau}p(\tau|s_0=s')\left(\sum_{t'=0}^{t-1}\gamma^{t'}r'_{t'}+\gamma^t U(s_t)-\hat{V}(s')\right) \tag{36}$$

$$=-2p(s_0=s')\left(\mathbb{E}\left[\sum_{t'=0}^{t-1}\gamma^{t'}r_{t'}+\gamma^t U(s_t)-\hat{V}(s_0)\,\middle|\,s_0=s'\right]\right)=0 \tag{37}$$

since $\mathbb{E}\left[\sum_{t'=0}^{t-1}\gamma^{t'}f_{t'}\,\middle|\,s_0=s'\right]=0$. If $p(s_0=s')>0$, then

$$\hat{V}(s') = \mathbb{E}\left[\sum_{t'=0}^{t-1}\gamma^{t'}r_{t'}+\gamma^t U(s_t)\,\middle|\,s_0=s'\right] \tag{38}$$

which completes the first part of the proof. Next, we prove the second part of the theorem regarding $f$. Similar to the proof of Theorem 1, we consider

$$\frac{\partial\mathcal{L}}{\partial f(s',a')}=-2\sum_{k=0}^{t-1}\gamma^k\underset{\tau\sim\pi}{\mathbb{E}}\left[\left(\sum_{t'=0}^{t-1}\gamma^{t'}r'_{t'}+\gamma^t U(s_t)-\hat{V}(s_0)\right)\left(\frac{\partial f_k}{\partial f(s',a')}\right)\right]+\lambda_{s'}\pi(a'|s')=0 \tag{39}$$

.

Following the proof of Theorem 1, we know that

$$\underset{\tau\sim\pi}{\mathbb{E}}\left[\left(\sum_{t'=0}^{t-1}\gamma^{t'}r'_{t'}+\gamma^t U(s_t)-\hat{V}(s_0)\right)\left(\frac{\partial f_k}{\partial f(s',a')}\right)\right] \tag{40}$$

$$=p(s_k=s',a_k=a')\underset{\tau\sim\pi}{\mathbb{E}}\left[\sum_{t'=0}^{t-1}\gamma^{t'}r'_{t'}+\gamma^t U(s_t)-\hat{V}(s_0)\,\middle|\,s_k=s',a_k=a'\right] \tag{41}$$

We substitute this back into Equation 39 and sum over the action space, which results in

$$\sum_{a'\in\mathcal{A}}\frac{\partial\mathcal{L}}{\partial f(s',a')}=-2\sum_{k=0}^{t-1}\gamma^k p(s_k=s')\underset{\tau\sim\pi}{\mathbb{E}}\left[\sum_{t'=0}^{t-1}\gamma^{t'}r'_{t'}+\gamma^t U(s_t)-\hat{V}(s_0)\,\middle|\,s_k=s'\right]+\lambda_{s'}=0 \tag{42}$$

In other words,

$$\lambda_{s'}=2\sum_{k=0}^{t-1}\gamma^k p(s_k=s')\underset{\tau\sim\pi}{\mathbb{E}}\left[\sum_{t'=0}^{t-1}\gamma^{t'}r'_{t'}+\gamma^t U(s_t)-\hat{V}(s_0)\,\middle|\,s_k=s'\right] \tag{43}$$

Substituting this back again into Equation 39, we have

$$\sum_{k=0}^{t-1}\gamma^k p(s_k=s',a_k=a')\left(\underset{\tau\sim\pi}{\mathbb{E}}\left[\sum_{t'=0}^{t-1}\gamma^{t'}r'_{t'}+\gamma^t U(s_t)-\hat{V}(s_0)\,\middle|\,s_k=s',a_k=a'\right]\right.$$
$$\left.-\underset{\tau\sim\pi}{\mathbb{E}}\left[\sum_{t'=0}^{t-1}\gamma^{t'}r'_{t'}+\gamma^t U(s_t)-\hat{V}(s_0)\,\middle|\,s_k=s'\right]\right)=0. \tag{44}$$

Using the Markov property, the term in the parentheses can be simplified to

$$\underset{\tau\sim\pi}{\mathbb{E}}\left[\sum_{t'=0}^{t-1}\gamma^{t'}r'_{t'}+\gamma^t U(s_t)-\hat{V}(s_0)\,\middle|\,s_k=s',a_k=a'\right] \tag{45}$$

$$-\underset{\tau\sim\pi}{\mathbb{E}}\left[\sum_{t'=0}^{t-1}\gamma^{t'}r'_{t'}+\gamma^t U(s_t)-\hat{V}(s_0)\,\middle|\,s_k=s'\right]$$

$$=\gamma^k\left(\underset{\tau\sim\pi}{\mathbb{E}}\left[\sum_{t'=0}^{t-k-1}\gamma^{t'}r_{t'}+\gamma^{t-k}U(s_{t-k})\,\middle|\,s_0=s',a_0=a'\right]\right.$$
$$\left.-\underset{\tau\sim\pi}{\mathbb{E}}\left[\sum_{t'=0}^{t-k-1}\gamma^{t'}r_{t'}+\gamma^{t-k}U(s_{t-k})\,\middle|\,s_0=s'\right]-f(s',a')\right) \tag{46}$$

Finally, if $(s',a')$ is reachable within $t-1$, then

$$f(s',a')=\frac{1}{\sum_{k=0}^{t-1}w_k(s')}\sum_{k=0}^{t-1}w_k(s')\left(\underset{\tau\sim\pi}{\mathbb{E}}\left[\sum_{t'=0}^{t-k-1}\gamma^{t'}r_{t'}+\gamma^{t-k}U(s_{t-k})\,\middle|\,s_0=s',a_0=a'\right]\right. \tag{47}$$

$$\left.-\underset{\tau\sim\pi}{\mathbb{E}}\left[\sum_{t'=0}^{t-k-1}\gamma^{t'}r_{t'}+\gamma^{t-k}U(s_{t-k})\,\middle|\,s_0=s'\right]\right), \tag{48}$$

with the shorthand $w_k(s')=\gamma^{2k}p(s_k=s')$. $\qquad\square$

# B    Integrating PPO with DAE

The PPO clipping loss is given by

$$L_\pi = \mathbb{E}\left[\min\left(\frac{\pi_\theta(a|s)}{\mu(a|s)}\hat{A}(s,a), \operatorname{clip}\left(\frac{\pi_\theta(a|s)}{\mu(a|s)}, 1-\epsilon, 1+\epsilon\right)\hat{A}(s,a)\right)\right], \tag{49}$$

where $\pi_\theta$ is the policy that is being optimized, $\mu$ is the sampling policy and $\hat{A}(s,a)$ is the estimated advantage function. In practice, an entropy loss $H_\pi = \sum_{a\in\mathcal{A}} \pi(a|s)\log\pi(a|s)$ is usually added to $L_\pi$ to encourage exploration. Note that, unlike the original PPO which samples mini-batches of frames, we sample on a trajectory-by-trajectory basis. For example, assume the batch size is 256 and $n = 128$ for the backup horizon, then each batch contains 2 128-step trajectories.

# C    Experiment details

## C.1    Computational resources

All the experiments were performed on an internal cluster of NVIDIA A100 GPUs. Training a MinAtar agent in a single environment takes less than 30 minutes (wall-clock time). Training an ALE agent on a single environment may take up to 9 hours (wall-clock time) depending on the network capacity.

## C.2    Variation study

We train a DAE baseline PPO agent with the hyperparameters described below and save the states encountered by each actor along with a checkpoint of the policy at the start of each PPO iteration for 50 iterations. Afterwards, we sample uniformly 1024 states from the set of collected states. For these states, we perform 128 MC rollouts (up to 512 steps) for every action (the MinAtar Breakout environment has 3 actions) to estimate the true $Q(s,a)$, $V(s)$ and $A(s,a)$ for every policy.

## C.3    Synthetic environment

To avoid giving DAE an unfair advantage because it could learn a function that ignores the input (since the advantage function is the same for each state), we randomly swap the rewards of u and d at each state at the beginning of the experiment.

We summarize the hyperparameters in Table 2. The policy and the value/advantage network are modeled separately. We parameterize the policy using $\theta_{ij} \in R^{128\times2}$, representing the logit of each action. $\theta_{ij}$ are initialized to 0 at the beginning of each experiment. The value/advantage network consists of MLPs of 2 hidden layers of size 256, and the input states are represented using one-hot encoding.

Table 2: Hyperparameters for the synthetic experiment.

| Parameter | Value GAE    DAE |
|---|---|
| Discount $\gamma$ | 0.99 |
| $N_{\text{iterations}}$ | 1000 |
| Optimizer | Adam |
| Learning rate (value/policy) | 0.001/0.01 |
| Adam $\beta$ | (0.9, 0.999) |
| Adam $\epsilon$ | $10^{-3}$ |
| Sample trajectories per iteration | 4 |
| Value gradients per iteration | 4 |
| Policy gradients per iteration | 1 |
| Batch Size | Whole dataset |
| GAE $\lambda$ | 0.95          — |

### C.4 MinAtar & ALE

#### C.4.1 Preprocessing

**MinAtar.** We turn off sticky action and difficulty ramping to further simplify the environments.

**ALE.** We follow the standard preprocessing procedures from Mnih et al. [2015], except for the max frames per episode where we used the default value from the gym implementation [Brockman et al., 2016]. See Table 3 for a summary of the parameters.

Table 3: ALE preprocessing parameters.

| Parameter | Value |
|---|---|
| Grey-scaling | True |
| Observation down-sampling | 84×84 |
| Frame stack | 4 |
| Frame skip | 4 |
| Reward clipping | $[-1, 1]$ |
| Terminal on loss of life | True |
| Max frames per episode | 400K |

#### C.4.2 Network architecture

For the MinAtar and the ALE experiments, we summarize the baseline network architectures in Figure 6. For the wide network experiments, we simply multiply the numbers of channels and widths of each hidden layer by 4 (MinAtar) or 2 (ALE). We follow Espeholt et al. [2018] for the Deep network architecture used in the ALE experiments, except we multiply the number of channels of each layer by 4 and increase the width of the last fully connected layer to 512. Additionally, we use SkipInit De and Smith [2020] to initialize the residual connections, which we found to stabilize learning.

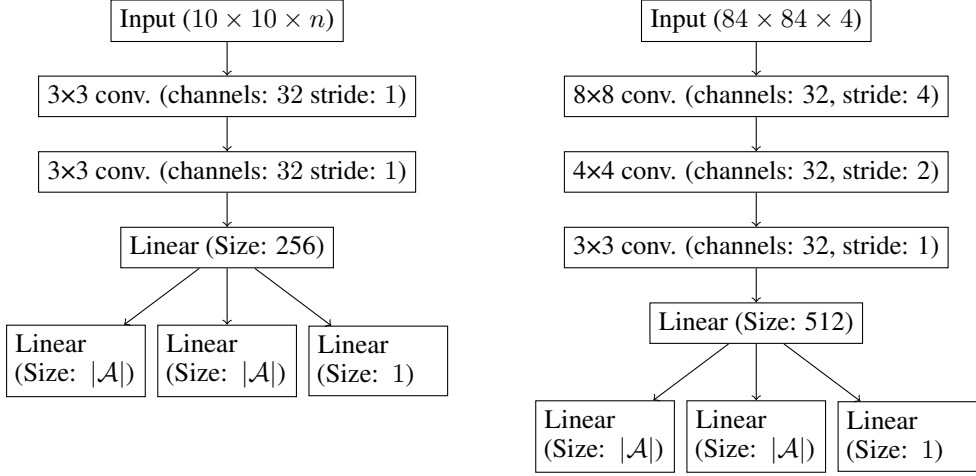

Figure 6: The baseline network architectures for the MinAtar experiments (left) and the ALE experiments (right). Each hidden layer is followed by a ReLU activation. The three output streams correspond to $A(s, a)$, $\pi(a|s)$, and $V(s)$, respectively.

#### C.4.3 PPO hyperparameters

We use the tuned PPO hyperparameters from Raffin [2020] for GAE. As for DAE, we additionally tune the number of epochs per iteration {4, 6, 8} and the $\beta_V$ coefficient {0.5, 1, 1.5} using the MinAtar environments. We also increase the number of parallel actors from 8 to 1024 to speed up training, the number 1024 was chosen to maximize GPU memory usage in the ALE experiments.

Among the tuned hyperparameters, we found that having a large number of parallel actors is the most important one. This is likely because DAE relies entirely on the network to approximate the advantage function, so having a huge batch of data at each PPO iteration is critical to having reliable estimates. Aside from the above-mentioned hyperparameters, we have also tried using separate networks for the policy and the advantage function, or replacing PPO clipping with KL-divergence penalties, but found them less effective than the original PPO algorithm.

We summarize the final set of hyperparameters in Table 4. The learning rate and the PPO clipping $\epsilon$ are linearly annealed towards 0 throughout training.

Table 4: Hyperparameters for the ALE experiments.

| Parameter | Value | |
|---|---|---|
| | GAE | DAE |
| Discount $\gamma$ | 0.99 | |
| $N_{\text{actor}}$ | 1024 | |
| $N_{\text{steps}}$ | 128 | |
| Optimizer | Adam | |
| Learning rate | 0.00025 | |
| Adam $\beta$ | (0.9, 0.999) | |
| Adam $\epsilon$ | $10^{-5}$ | |
| $N_{\text{Epochs}}$ | 4 | 6 |
| Batch Size | 256 | |
| PPO Clipping $\epsilon$ | 0.1 | |
| $\beta_V$ | 0.5 | 1.5 |
| $\beta_{\text{ent}}$ | 0.01 | |
| GAE $\lambda$ | 0.95 | — |
| Weight Initialization | orthogonal | |

# D  Ablation study

In addition to the GAE baseline, here we consider two additional baselines to demonstrate the effectiveness of DAE.

1. Indirect: We learn both $Q$ and $V$ separately by minimizing the $n$-step bootstrapping losses.

$$L_Q = \mathbb{E}\left[ \left(\hat{Q}(s,a) - \left(r_0 + ... + \gamma^{n-1}r_{n-1} + \gamma^n V_{\text{target}}(s_n)\right)\right)^2 \middle| s_0{=}s, a_0 = a\right] \quad (50)$$

$$L_V = \mathbb{E}\left[ \left(\hat{V}(s) - \left(r_0 + ... + \gamma^{n-1}r_{n-1} + \gamma^n V_{\text{target}}(s_n)\right)\right)^2 \middle| s_0{=}s\right] \quad (51)$$

   where $\hat{Q}$ and $\hat{V}$ are the learned $Q$-function and the value function. $\hat{Q}$ and $\hat{V}$ can then be used to estimate the advantage function via $\hat{A}(s,a) = \hat{Q}(s,a) - \hat{V}(s)$. This baseline demonstrates the effectiveness of learning the advantage function directly.

2. Duel: Based on Wang et al. [2016b], we slightly modify the original loss function to extend to the $n$-step setting. The new loss function is now

$$L = \mathbb{E}\left[ \left(\hat{V}(s) + \hat{A}(s,a) - \left(r_0 + ... + \gamma^{n-1}r_{n-1} + \gamma^n V_{\text{target}}(s_n)\right)\right)^2 \middle| s_0{=}s, a_0 = a\right], \quad (52)$$

   where $\hat{V}$ and $\hat{A}$ are being learned. Essentially, this differs from the DAE loss in whether we sum over the advantage function over time steps. Unlike the original dueling architecture, we use the learned policy instead of the uniform policy to enforce the $\pi$-centered constraint. This baseline demonstrates the effectiveness of the DAE loss (sum over advantage functions).

We compare their performance in policy optimization using the MinAtar suite with the settings and DAE hyperparameters (baseline network) described in Appendix C. We show the learning curves for individual environments and the normalized curves in Figure 7 and Figure 8. Our results show that the Indirect method performs worse than both DAE and Duel, suggesting that explicitly modelling the advantage function can be beneficial. Furthermore, by comparing the results from DAE and Duel, we see substantial gains by switching to the DAE loss, even surpassing the performance of GAE. This suggests that jointly estimating the advantage function across time steps is crucial.

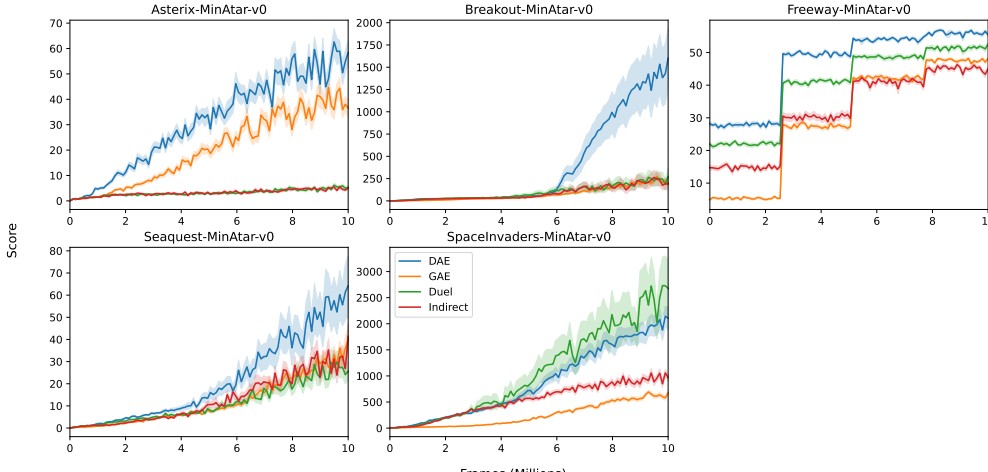

Figure 7: Learning curves for the raw scores in the MinAtar experiments. Lines and shadings represent the average and one standard error over 50 runs.

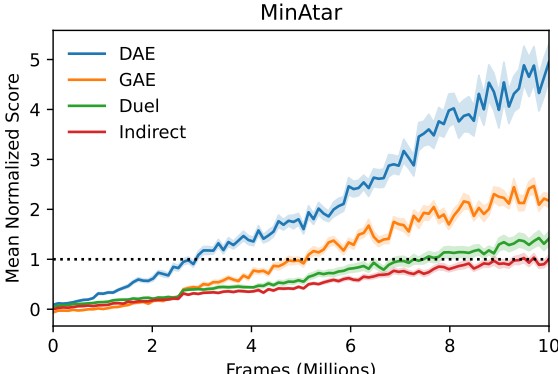

Figure 8: Normalized learning curves for the ablation study. Scores are normalized independently for each environment based on scores from the Indirect baseline before aggregated (50 seeds).

# E   Additional results

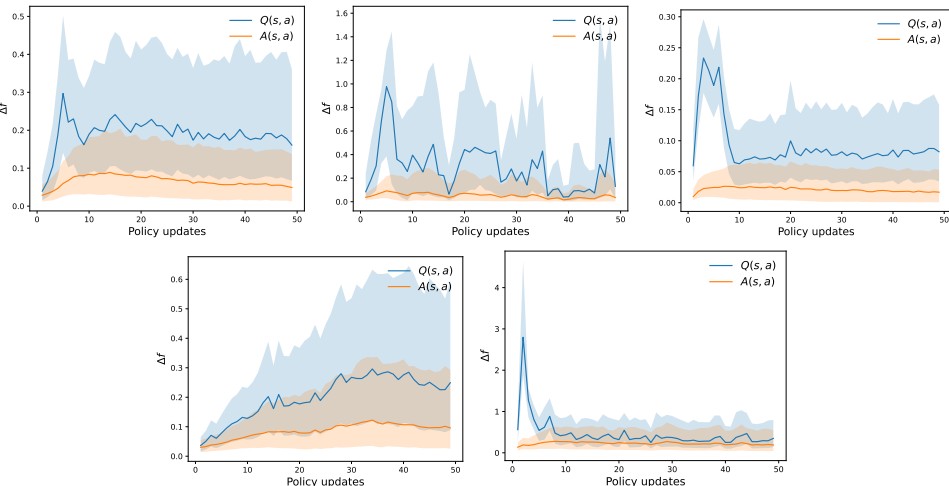

Figure 9: Variations of the advantage function and the Q-function from the MinAtar environments (top-left to bottom-right are Asterix, Breakout, Freeway, Seaquest and Space Invaders, respectively). See Appendix C for details.

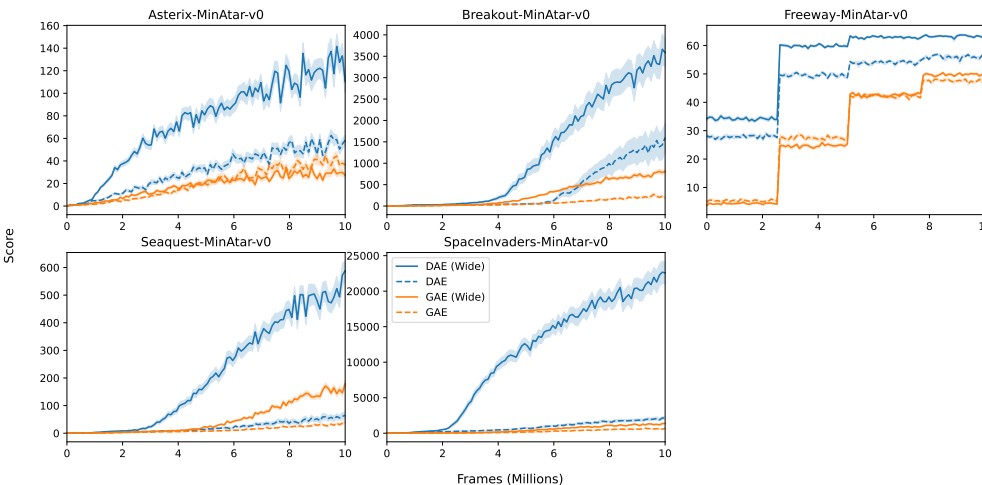

Figure 10: Learning curves for the raw scores in the MinAtar experiments. Lines and shadings represent the average and one standard error over 50 runs.

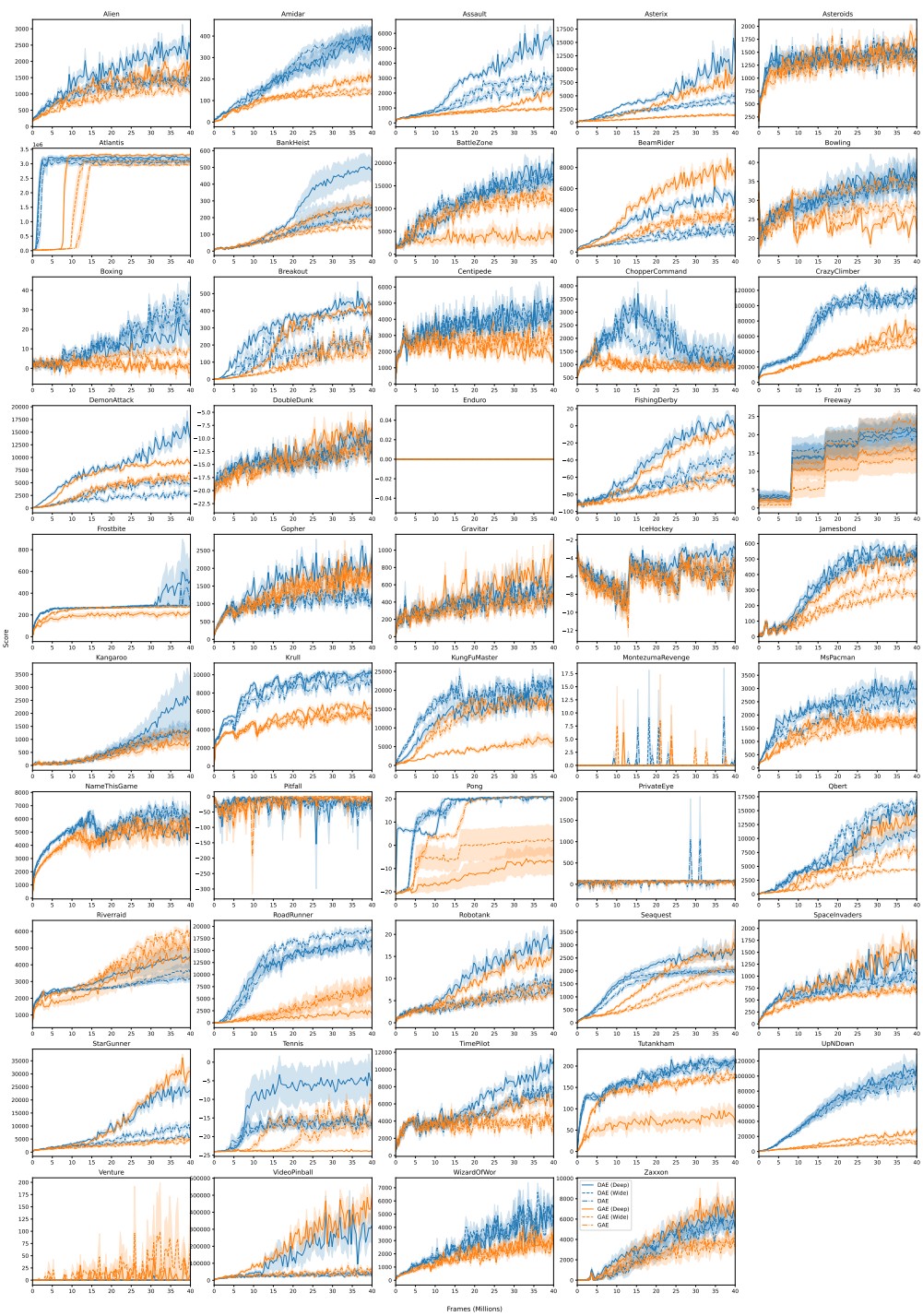

Figure 11: Learning curves for the raw scores in the ALE experiments. Lines and shadings represent the average and one standard error over 10 runs.

Table 5: Overall and last scores (defined in Section 5) on the MinAtar environments with baseline network. Numbers represent (mean)±(1 standard error of the mean).

| Environment | Metric | | | |
| | Overall | | Last | |
| | GAE | DAE | GAE | DAE |
|---|---|---|---|---|
| Asterix | 6.6±0.1 | 10.5±0.1 | 27.4±0.6 | 40.7±1.0 |
| Breakout | 10.0±0.1 | 17.2±0.3 | 168.2±11.3 | 844.7±152.3 |
| Freeway | 30.2±0.3 | 46.5±0.2 | 47.5±0.2 | 55.9±0.2 |
| Seaquest | 4.4±0.1 | 6.3±0.3 | 26.4±1.9 | 45.9±7.2 |
| SpaceInvaders | 43.7±0.3 | 143.7±2.6 | 504.2±10.3 | 1675.3±140.1 |

Table 6: Overall and last scores (defined in Section 5) on the MinAtar environments with wide network. Numbers represent (mean)±(1 standard error of the mean).

| Environment | Metric | | | |
| | Overall | | Last | |
| | GAE | DAE | GAE | DAE |
|---|---|---|---|---|
| Asterix | 7.3±0.1 | 18.0±0.1 | 22.1±0.4 | 89.3±1.9 |
| Breakout | 12.0±0.0 | 23.3±0.4 | 637.7±21.3 | 2409.5±190.5 |
| Freeway | 29.9±0.2 | 54.8±0.1 | 49.8±0.2 | 63.4±0.1 |
| Seaquest | 7.0±0.2 | 16.4±0.5 | 123.7±2.9 | 442.1±22.6 |
| SpaceInvaders | 62.6±0.4 | 276.0±2.0 | 1058.0±18.1 | 11743.3±315.9 |

Table 7: Overall and last scores (defined in Section 5) on the ALE environments with baseline network. Numbers represent (mean)±(1 standard error of the mean).

| Environment | Metric | | | |
| --- | --- | --- | --- | --- |
| | Overall | | Last | |
| | GAE | DAE | GAE | DAE |
| Alien | 809.9±9.5 | 1042.8±36.3 | 1165.7±24.3 | 1372.7±109.1 |
| Amidar | 90.5±0.9 | 201.8±13.4 | 129.4±3.6 | 394.6±38.6 |
| Assault | 653.1±4.0 | 1026.4±42.3 | 926.8±12.5 | 2205.1±115.4 |
| Asterix | 724.8±4.0 | 1696.8±51.7 | 1368.6±11.1 | 3750.1±168.4 |
| Asteroids | 1220.7±5.6 | 1235.1±11.6 | 1395.7±15.2 | 1392.3±19.4 |
| Atlantis | 161787.1±3594.5 | 886857.2±54327.4 | 2039346.4±30987.7 | 2888011.1±68326.9 |
| BankHeist | 82.1±2.0 | 111.1±15.3 | 144.2±4.6 | 257.8±61.0 |
| BattleZone | 9286.7±219.8 | 11440.0±358.6 | 12813.0±389.6 | 16302.0±628.5 |
| BeamRider | 1617.9±11.8 | 1027.0±33.1 | 2968.8±36.0 | 1729.9±56.2 |
| Bowling | 31.6±0.5 | 30.7±1.6 | 34.7±1.2 | 36.1±2.9 |
| Boxing | 5.7±0.2 | 13.8±1.8 | 8.5±0.3 | 25.9±3.1 |
| Breakout | 46.8±1.2 | 97.3±4.4 | 160.3±3.2 | 234.9±8.6 |
| Centipede | 2566.4±13.7 | 3238.3±76.3 | 2682.3±57.1 | 3915.8±222.1 |
| ChopperCommand | 1137.9±14.5 | 1772.5±108.7 | 1073.4±19.2 | 1587.3±137.8 |
| CrazyClimber | 33832.7±493.6 | 72051.1±2153.6 | 55877.1±1514.7 | 112319.5±1925.6 |
| DemonAttack | 2334.5±42.5 | 1373.5±46.2 | 5736.1±118.9 | 2477.2±108.6 |
| DoubleDunk | -14.8±0.1 | -15.5±0.6 | -12.1±0.3 | -12.5±1.0 |
| Enduro | 0.0±0.0 | 0.0±0.0 | 0.0±0.0 | 0.0±0.0 |
| FishingDerby | -79.8±0.3 | -72.6±1.3 | -68.6±0.8 | -60.4±2.3 |
| Freeway | 15.0±2.5 | 14.2±3.2 | 23.4±2.2 | 19.5±4.4 |
| Frostbite | 241.2±0.4 | 263.4±5.3 | 272.0±1.1 | 367.9±86.6 |
| Gopher | 1163.2±56.4 | 910.3±13.3 | 1666.0±90.5 | 1137.2±49.4 |
| Gravitar | 310.9±12.6 | 347.3±10.4 | 373.7±26.3 | 443.5±15.8 |
| IceHockey | -5.9±0.1 | -6.1±0.1 | -4.5±0.1 | -5.1±0.2 |
| Jamesbond | 138.1±6.2 | 237.4±6.2 | 277.1±13.2 | 507.8±6.9 |
| Kangaroo | 450.5±36.2 | 503.2±124.4 | 1022.0±97.2 | 1331.0±334.6 |
| Krull | 4573.8±35.9 | 7496.0±242.4 | 5642.2±72.8 | 9034.0±246.0 |
| KungFuMaster | 10614.3±124.6 | 14286.3±439.1 | 17389.1±250.8 | 20535.3±911.5 |
| MontezumaRevenge | 0.0±0.0 | 0.1±0.0 | 0.2±0.2 | 0.1±0.1 |
| MsPacman | 1148.4±18.4 | 1714.7±81.2 | 1674.8±20.2 | 2501.0±189.7 |
| NameThisGame | 4427.7±23.4 | 5013.9±64.4 | 5685.5±59.2 | 6016.3±91.8 |
| Pitfall | -4.5±0.4 | -7.5±0.6 | -0.1±0.1 | -12.0±6.7 |
| Pong | 8.9±0.2 | 12.5±0.5 | 20.9±0.0 | 20.7±0.1 |
| PrivateEye | 74.8±11.0 | 87.3±12.0 | 78.2±13.1 | 86.0±6.2 |
| Qbert | 1987.7±22.3 | 4622.6±250.9 | 4357.4±41.0 | 11119.6±1102.2 |
| Riverraid | 3347.2±39.2 | 2672.0±53.0 | 5133.2±93.0 | 3150.8±172.1 |
| RoadRunner | 2133.9±801.5 | 7445.8±572.7 | 6530.9±2194.0 | 16146.3±968.6 |
| Robotank | 3.9±0.1 | 4.1±0.3 | 6.0±0.2 | 6.9±0.9 |
| Seaquest | 669.0±26.9 | 1212.0±66.8 | 1580.4±68.8 | 2049.8±135.9 |
| SpaceInvaders | 474.1±1.5 | 614.6±29.4 | 665.4±8.4 | 914.9±68.6 |
| StarGunner | 2677.5±30.3 | 3127.2±103.1 | 4491.8±143.1 | 5849.2±232.6 |
| Tennis | -23.1±0.1 | -23.1±0.1 | -16.6±0.2 | -17.6±0.5 |
| TimePilot | 3417.1±23.9 | 4733.7±96.6 | 3758.4±75.5 | 7252.7±373.1 |
| Tutankham | 119.3±2.3 | 150.2±5.0 | 172.5±2.4 | 196.0±8.3 |
| UpNDown | 5714.1±89.4 | 19671.7±707.7 | 14288.2±516.8 | 85438.4±6152.3 |
| Venture | 0.1±0.1 | 0.0±0.0 | 0.0±0.0 | 0.0±0.0 |
| VideoPinball | 22791.1±208.0 | 20279.4±627.9 | 28256.7±1251.8 | 23958.6±1357.6 |
| WizardOfWor | 1605.4±33.7 | 2414.4±111.1 | 2489.5±110.4 | 4161.3±216.0 |
| Zaxxon | 1749.7±171.1 | 2520.5±339.9 | 3897.9±192.7 | 5612.2±540.0 |

Table 8: Overall and last scores (defined in Section 5) on the ALE environments with Wide network. Numbers represent (mean)±(1 standard error of the mean).

| Environment | Metric | | | |
| | Overall | | Last | |
| | GAE | DAE | GAE | DAE |
|---|---|---|---|---|
| Alien | 957.5±14.4 | 1172.2±50.8 | 1452.3±39.5 | 1461.5±110.4 |
| Amidar | 99.1±1.3 | 191.3±18.0 | 150.2±5.8 | 391.8±46.3 |
| Assault | 648.3±6.0 | 1195.7±40.3 | 991.1±21.1 | 3002.4±78.2 |
| Asterix | 740.8±8.7 | 1960.4±77.1 | 1427.5±25.8 | 4782.3±516.5 |
| Asteroids | 1096.7±11.9 | 1333.9±14.2 | 1278.2±21.3 | 1549.0±18.7 |
| Atlantis | 187676.0±3832.8 | 1074670.4±53480.9 | 2261050.5±32146.1 | 3006634.5±63364.8 |
| BankHeist | 98.2±2.6 | 105.3±3.9 | 188.0±6.1 | 214.6±11.9 |
| BattleZone | 8959.6±211.3 | 10820.1±611.7 | 12155.0±285.5 | 15903.0±661.6 |
| BeamRider | 1839.0±16.4 | 1275.4±27.3 | 3540.0±23.2 | 2203.2±76.7 |
| Bowling | 26.3±0.5 | 29.9±1.9 | 28.5±0.7 | 34.4±2.6 |
| Boxing | 1.6±0.1 | 15.8±1.9 | 1.9±0.1 | 34.3±4.7 |
| Breakout | 62.4±2.3 | 155.3±2.1 | 217.4±8.4 | 383.4±4.1 |
| Centipede | 2426.7±24.3 | 3394.7±85.5 | 2482.7±45.4 | 4517.1±286.8 |
| ChopperCommand | 1094.5±40.5 | 2264.5±158.6 | 1024.7±33.5 | 2261.3±282.9 |
| CrazyClimber | 31296.7±476.3 | 77555.5±1712.7 | 49524.3±1617.6 | 118288.9±1709.6 |
| DemonAttack | 2452.2±33.6 | 2226.0±69.7 | 5902.1±78.8 | 4893.3±289.9 |
| DoubleDunk | -15.4±0.1 | -16.1±0.3 | -12.8±0.3 | -14.4±0.7 |
| Enduro | 0.0±0.0 | 0.0±0.0 | 0.0±0.0 | 0.0±0.0 |
| FishingDerby | -75.3±1.0 | -64.1±3.3 | -52.9±2.1 | -35.6±7.1 |
| Freeway | 8.2±2.3 | 15.6±2.9 | 13.4±3.7 | 21.8±3.8 |
| Frostbite | 246.1±0.8 | 262.5±2.4 | 277.1±2.4 | 284.6±6.3 |
| Gopher | 1159.6±54.1 | 980.6±31.5 | 1591.8±90.1 | 1225.9±102.2 |
| Gravitar | 327.9±18.6 | 378.5±15.3 | 450.6±35.5 | 524.8±30.9 |
| IceHockey | -6.2±0.1 | -5.8±0.1 | -5.3±0.1 | -4.5±0.1 |
| Jamesbond | 168.1±5.6 | 235.1±10.0 | 385.9±10.1 | 506.6±9.3 |
| Kangaroo | 533.7±32.7 | 474.2±85.9 | 1303.6±67.8 | 1346.8±248.9 |
| Krull | 4544.6±45.5 | 8340.4±58.4 | 5361.6±69.8 | 9953.9±81.8 |
| KungFuMaster | 9384.7±404.9 | 14780.2±424.2 | 15674.3±350.1 | 21483.3±658.4 |
| MontezumaRevenge | 0.1±0.0 | 0.1±0.0 | 0.3±0.2 | 0.0±0.0 |
| MsPacman | 1338.9±9.3 | 2166.2±138.8 | 1828.2±17.9 | 3065.3±278.9 |
| NameThisGame | 4297.1±22.8 | 5183.5±98.8 | 5303.2±97.2 | 6250.2±171.7 |
| Pitfall | -4.7±0.4 | -9.0±1.1 | 0.0±0.0 | -5.7±2.2 |
| Pong | -4.3±5.1 | 12.5±0.5 | 2.3±6.3 | 20.7±0.1 |
| PrivateEye | 44.5±13.2 | 48.2±11.4 | 61.6±11.7 | 69.6±8.8 |
| Qbert | 2945.0±37.0 | 6190.1±156.5 | 7748.5±182.2 | 16239.6±361.3 |
| Riverraid | 3623.1±65.2 | 2723.5±78.2 | 5830.5±116.7 | 3630.6±379.1 |
| RoadRunner | 2006.4±631.7 | 9600.6±406.5 | 6461.3±1517.3 | 18767.2±484.3 |
| Robotank | 4.7±0.1 | 4.8±0.3 | 7.5±0.2 | 9.1±0.9 |
| Seaquest | 793.3±21.5 | 1285.6±49.1 | 2100.6±79.6 | 1967.2±126.8 |
| SpaceInvaders | 485.1±1.3 | 658.5±28.0 | 699.9±11.7 | 1098.3±73.4 |
| StarGunner | 2319.1±32.2 | 4286.1±195.1 | 3752.2±121.5 | 9123.1±510.7 |
| Tennis | -23.7±0.0 | -23.2±0.1 | -20.4±0.9 | -17.6±0.8 |
| TimePilot | 3514.2±17.4 | 4813.9±67.7 | 3902.5±58.4 | 7649.6±288.4 |
| Tutankham | 123.7±1.5 | 159.8±2.7 | 180.5±2.2 | 209.0±4.3 |
| UpNDown | 4968.0±70.5 | 20478.4±881.1 | 11014.5±246.0 | 94550.9±6967.9 |
| Venture | 21.8±21.7 | 0.1±0.1 | 42.4±42.4 | 0.2±0.2 |
| VideoPinball | 30055.3±555.4 | 24036.4±704.1 | 46653.9±2099.4 | 32161.8±1625.1 |
| WizardOfWor | 1622.4±37.2 | 2665.1±77.0 | 2591.7±100.3 | 5057.0±244.2 |
| Zaxxon | 1661.7±425.7 | 2048.2±395.2 | 3496.6±720.5 | 5275.2±423.5 |

Table 9: Overall and last scores (defined in Section 5) on the ALE environments with Deep network. Numbers represent (mean)±(1 standard error of the mean).

| Environment | Metric | | | |
|---|---|---|---|---|
| | Overall | | Last | |
| | GAE | DAE | GAE | DAE |
| Alien | 1151.3±9.4 | 1501.4±59.5 | 1735.9±42.2 | 2396.3±138.1 |
| Amidar | 115.4±3.8 | 192.7±22.8 | 205.8±15.5 | 362.0±55.2 |
| Assault | 788.5±8.7 | 1587.8±48.8 | 1862.4±81.2 | 4963.1±280.4 |
| Asterix | 2010.7±58.3 | 3403.3±80.7 | 7239.5±498.9 | 11861.7±1190.4 |
| Asteroids | 1309.5±8.3 | 1335.8±14.6 | 1581.1±27.7 | 1531.2±28.1 |
| Atlantis | 250696.5±3918.9 | 1220135.9±62706.7 | 2649450.3±24887.5 | 3082321.8±64162.3 |
| BankHeist | 137.3±14.4 | 180.4±11.8 | 279.2±29.0 | 494.1±74.9 |
| BattleZone | 3373.1±872.2 | 10584.5±775.1 | 4180.0±1298.8 | 18410.0±1261.4 |
| BeamRider | 3578.3±39.5 | 2801.9±66.4 | 7597.5±119.8 | 5179.8±182.5 |
| Bowling | 23.9±0.2 | 31.9±2.2 | 24.2±0.2 | 37.1±3.3 |
| Boxing | 1.5±0.0 | 10.0±3.5 | 1.7±0.1 | 19.3±6.6 |
| Breakout | 108.2±4.3 | 193.5±3.4 | 410.2±6.6 | 440.6±6.2 |
| Centipede | 2048.1±17.2 | 3407.9±89.5 | 1897.2±28.4 | 4505.4±168.7 |
| ChopperCommand | 1006.2±23.7 | 2177.2±113.3 | 973.0±28.0 | 2135.1±200.6 |
| CrazyClimber | 37238.2±1126.0 | 71207.8±3992.4 | 66005.2±3164.9 | 109013.9±4292.6 |
| DemonAttack | 4392.1±59.0 | 5906.1±198.6 | 9064.1±95.4 | 13841.7±922.5 |
| DoubleDunk | -13.7±0.3 | -15.0±0.8 | -8.5±0.7 | -12.7±1.4 |
| Enduro | 0.0±0.0 | 0.0±0.0 | 0.0±0.0 | 0.0±0.0 |
| FishingDerby | -49.2±1.1 | -36.1±3.9 | -8.7±1.6 | 8.4±3.7 |
| Freeway | 11.2±2.2 | 14.5±2.8 | 16.0±2.5 | 20.9±3.7 |
| Frostbite | 184.8±18.7 | 283.9±22.5 | 207.9±23.0 | 494.1±200.5 |
| Gopher | 1222.7±73.2 | 1444.7±103.7 | 1721.5±137.2 | 2000.1±143.8 |
| Gravitar | 482.7±32.6 | 397.5±18.5 | 710.5±61.9 | 581.8±41.3 |
| IceHockey | -6.5±0.0 | -4.8±0.3 | -6.0±0.1 | -3.0±0.4 |
| Jamesbond | 230.5±12.9 | 267.5±9.3 | 489.4±16.3 | 530.4±12.2 |
| Kangaroo | 351.5±64.0 | 585.4±156.3 | 879.4±170.2 | 2453.9±899.2 |
| Krull | 5121.2±51.3 | 8464.9±48.6 | 6210.1±101.9 | 9847.3±95.1 |
| KungFuMaster | 3468.0±187.2 | 10871.2±999.5 | 6535.3±1089.9 | 16581.3±1497.8 |
| MontezumaRevenge | 0.0±0.0 | 0.1±0.0 | 0.0±0.0 | 0.4±0.4 |
| MsPacman | 1386.4±142.1 | 2150.3±72.3 | 1774.8±194.8 | 3057.2±219.6 |
| NameThisGame | 4101.7±458.2 | 4654.8±103.7 | 4757.8±577.4 | 5109.2±164.8 |
| Pitfall | -10.3±1.2 | -8.2±0.4 | -6.3±1.3 | -6.3±1.0 |
| Pong | -14.3±3.5 | 17.3±0.1 | -6.6±5.7 | 20.9±0.0 |
| PrivateEye | 53.2±0.8 | 53.7±5.1 | 54.0±3.2 | 55.0±6.3 |
| Qbert | 4165.5±414.2 | 5563.4±197.3 | 13009.7±1212.3 | 15177.2±399.3 |
| Riverraid | 2740.5±542.5 | 3206.6±212.4 | 4392.3±1105.7 | 4455.4±610.3 |
| RoadRunner | 867.4±448.3 | 8604.9±1041.4 | 2153.5±1231.5 | 16665.6±795.0 |
| Robotank | 7.4±0.1 | 8.7±0.7 | 14.0±0.3 | 17.0±1.2 |
| Seaquest | 1111.5±42.7 | 1553.3±64.7 | 2776.3±249.5 | 2723.7±287.0 |
| SpaceInvaders | 755.2±9.6 | 755.8±20.9 | 1421.9±58.5 | 1269.5±75.3 |
| StarGunner | 7281.1±222.2 | 7400.8±253.4 | 29961.4±1274.0 | 23087.6±1421.8 |
| Tennis | -23.9±0.0 | -16.3±3.4 | -23.9±0.0 | -6.6±5.4 |
| TimePilot | 4548.1±143.9 | 5942.2±156.1 | 6523.9±377.6 | 10510.2±313.1 |
| Tutankham | 57.6±12.9 | 160.8±3.4 | 85.9±19.1 | 212.5±6.5 |
| UpNDown | 7692.6±254.0 | 23790.1±2304.3 | 22454.7±1222.5 | 101203.3±12747.0 |
| Venture | 6.9±4.6 | 0.0±0.0 | 12.1±8.1 | 0.0±0.0 |
| VideoPinball | 133908.8±4973.8 | 66261.9±10063.3 | 348512.7±23846.3 | 234364.9±55812.2 |
| WizardOfWor | 1641.1±107.4 | 2537.6±68.1 | 2599.3±225.9 | 4537.2±178.4 |
| Zaxxon | 3101.4±383.3 | 3011.4±509.4 | 6982.3±695.0 | 6438.6±1042.5 |