# OpenReview forum: "Direct Advantage Estimation"
_NeurIPS.cc/2022/Conference — NeurIPS 2022 Accept_

### Official Review · Reviewer_c2up · 2022-07-08

**Rating:** 8
**Confidence:** 3
**Soundness:** 4 excellent
**Presentation:** 3 good
**Contribution:** 4 excellent

**Summary:**

The authors propose a new method for estimating the advantage function of a policy that does not require first estimating the Q or value function. They demonstrate that on a variety of tasks, it works better than the "industry-standard" GAE.

**Questions:**

(1) Much of the motivation in the intro of the paper centers around being able to directly estimate the advantage function without having to first fit a value function. However, it seems like from Thm. 2 and beyond, the authors actually end up fitting a value function. Why does your proposed method not suffer from the same issues that plague value-based methods that you point out prior (e.g. small policy changes leading to large distribution shifts and inaccurate value function estimates)?

(2) You mention in the appendix that for DAE, one needs to sample entire trajectories rather than mini-batches. Do you have a wall-clock time comparison between DAE and GAE?

(3) The concern about estimating the sum of advantages for normalization reminds me a lot of the difficulties estimating the partition function for maximum entropy / energy-based / exponential family models. I wonder if you could borrow from the rich literature there to solve your problem.

(4) The DAE-Wide results are particularly good. Could you provide some reasons as to why this might be the case?

**Limitations:**

The authors correctly acknowledge that their method is dependent on discrete action spaces for enforcing the normalization constraint and provide some potential solutions. I think the related work section could be improved as well (i.e. there are other advantage estimation methods outside of GAE).

**Strengths And Weaknesses:**

(+) This paper was a joy to read. I appreciated the nuance of the discussion (e.g. the examples of where the advantage fn. was more or less policy dependent, the example of why TD-based methods can't work for advantage functions in general). The paper felt fundamental yet accessible, which is a hard balance to strike.

(+) The experiments were well-done and involved results averaged across many environments, indicating a significant gain in performance when switching from GAE to DAE.

(+) I appreciated the connection to causal effects and especially the Halpern citation :).

(-) In general, making decisions based on expected returns (rather than greedily) is a good idea. Some parts of the first few pages could be read as indicating otherwise. If I were you, I would focus on why advantages are good (cite PPO/Kakade&Langford) and delve into why (they are sort of an appropriate level of "local"), rather than discussing how Q functions depend on the policy at future steps (which is somewhat obvious and useful).

---

> ### Author Response · Authors · 2022-08-02
> **Response to Reviewer c2up**
>
> Thank you for your constructive and motivating feedback. We are glad you found the paper interesting.
>
> > Much of the motivation in the intro of the paper centers around being able to directly estimate the advantage function without having to first fit a value function. However, it seems like from Thm. 2 and beyond, the authors actually end up fitting a value function. Why does your proposed method not suffer from the same issues that plague value-based methods that you point out prior (e.g. small policy changes leading to large distribution shifts and inaccurate value function estimates)?
>
> While the bootstrapped method (Theorem 2) also suffers from the same issues mentioned in the motivation, we argue that this can be mitigated by using long horizon multistep backups when actions mostly have short-term effects. For example, consider the idealized case where the influence of an action persists only up to $m$ steps (i.e., the future state distribution is left unchanged by the action after $m$ steps), and a backup horizon of $n > m$. Then Theorem 2 suggests that the estimated advantage function is truncated after the first $m$ steps, meaning that $V_\text{target}$ would have no effect on the estimated advantage function.
>
> > You mention in the appendix that for DAE, one needs to sample entire trajectories rather than mini-batches. Do you have a wall-clock time comparison between DAE and GAE?
>
> The difference to the sampling process in PPO with GAE is that we require the samples to be sequences $(s_t, a_t, r_t, s_{t+1}, a_{t+1}, r_{t+1}, …)$ that are contiguous in time (non-terminated trajectories) to compute the loss. For example, when we say the batch size is 256 and the n-step is 128, we sample in each gradient step 2 sequences of 128-step transitions. Essentially, the only difference in wall-clock time comes from the computation of different loss functions, which we found to be negligible.
>
> > The concern about estimating the sum of advantages for normalization reminds me a lot of the difficulties estimating the partition function for maximum entropy / energy-based / exponential family models. I wonder if you could borrow from the rich literature there to solve your problem.
>
> Thank you very much for this suggestion. We agree that this looks like a promising direction for future work.
>
> > The DAE-Wide results are particularly good. Could you provide some reasons as to why this might be the case?
>
> We believe that this is mainly due to the way the advantage function is represented. Unlike GAE, which combines unbiased n-step rewards and the approximated value function to estimate the advantage function, DAE relies solely on the function approximator to represent the advantage function. This suggests that DAE should be able to make better use of the capacity of the function approximator than GAE, and our experiments further strengthen this point by showing that using a larger model can significantly improve the performance of DAE.

---

> > ### Comment · Reviewer_c2up · 2022-08-05
> > **Re:**
> >
> > Thanks for the thorough response. I think I understand what you said for (2)-(4). For (1), is your point that because the advantage function doesn't depend on the distant future, the variance of V_target (which scales with the horizon) doesn't end up hurting the estimation procedure much?

---

> > > ### Author Response · Authors · 2022-08-05
> > > **Re:**
> > >
> > > That's correct. More precisely, if $p(s_{t+n}|s_t, a_t)\approx p(s_{t+n}|s_t)$ (which we assume to be true when $n$ is large), then $E[V_\text{target}(s_{t+n})|s_t, a_t]\approx E[V_\text{target}(s_{t+n})|s_t]$. Consequently, the advantage estimate (Eq. 16) would also depend weakly on $V_\text{target}$.

---

> > > > ### Comment · Reviewer_c2up · 2022-08-05
> > > > **Re:**
> > > >
> > > > Got it, makes sense -- the softer version of your assumption would be a mixing time assumption (in the Markov chain sense) or an exponential stability assumption (in the linear controls sense) -- I might mention these explicitly. I'd be happy to bump my score up to an 8 if you'd be able to add discussion of this point + more discussion of the DAE-wide results.

---

> > > > > ### Author Response · Authors · 2022-08-09
> > > > > **Re:**
> > > > >
> > > > > Thank you for the suggestion, we've expanded the discussion on the softer condition (line 160-166) and the results on network capacity (line 306-314).

---

> > > > > > ### Comment · Reviewer_c2up · 2022-08-09
> > > > > > **Re:**
> > > > > >
> > > > > > Thanks! Bumped up my score.

---

### Official Review · Reviewer_9WbF · 2022-07-08

**Rating:** 7
**Confidence:** 4
**Soundness:** 3 good
**Presentation:** 2 fair
**Contribution:** 4 excellent

**Summary:**

The paper considers the problem of learning the advantage function directly (with no need for learning Q first). The authors coined the term $\pi$-normalized [a bit weird terminology] as any function whose expectation over $\pi$ is zero. Advantage function is an obvious example. Clearly, if the reward is shifted by any $\pi$-normalized function, the expected return remains unchanged. This property is then used to minimize the variance of return by making it disentangled from the policy. One main result (via Theorem 1) is that the optimal $\pi$-normalized function (in terms of minimizing the variance) to be subtracted from the reward is indeed the advantage function of the current policy for on-policy learning. This also gives rise to a learning paradigm with the loss function coming directly from the grounded minimization result in Theorem 1. This is then extended to N-step learning using the value function. Finally, promising experimental results are provided to confirm the method.

**Questions:**

- L54: Your definition of reward is indeed *expected* reward (expectation taken oven next state $s’$).

- Paragraph of L150: I generally accept that A is lower variance than Q but using PPO to demonstrate this is not adequate. In particular, I would examine this with a value-based method that directly reinforces Q, instead of AC methods, which normally does not give rise to accurate value function.

- The term $\pi$-normalized is somehow misleading, as normalization is a process of mapping to one, not zero. Consider a different term; perhaps something like  $\pi$-neutralized or $\pi$-nullified.

- From proof of theorem 1 (L473), it looks like that non-zero policy is a *missing* assumption, $\pi(s,a)\neq 0$ for all $(s,a)$. This kind of assumption is very normal in policy gradient results. I do not consider it as a downside.

- Using value-function for the N-step case is somehow defeating the purpose as the main premise is to disentangle advantage learning from value learning. Isn't it?

- The last part of the theory (from L232 onwards) is not well-written and requires more explanation and expansion. I recommend removing several parts from the first 4 pages (which are simply reviewing textbook stuff) and expand on the last part of the theory with more scrutiny. In particular, $V_{target}$ is not clearly defined and its dependence on learning $V$ is not clear. The trajectory augmentation seems also to be required. From the text, it is very hard to come up with a clear algorithmic machinery. I strongly suggest to add a algorithm block and summarize the details.


**Limitations:**

No limitation is explicitly mentioned, though from the future work discussion there are some comprehendible limitations:

- Action space is discrete.
- On-policy learning.

**Strengths And Weaknesses:**

Strengths:

- I liked the theoretical results. Direct learning of advantage function is novel and useful. Additionally, the minimization formulation from Theorem 1 is also interesting.

- Disentanglement of learning from current policy is indeed interesting, as it is also evident from the experimental results.

Weaknesses:

- The implementation of this paper requires several steps, which are somehow distributed in various parts of the paper. A clear algorithm block is missing.

- The second part of the theory is not presented clearly, and is too condensed.

- Minor: I do not understand the emphasis on some "causal" concepts. These are neither needed (or even used) nor are they any addition to the presented theory.

---

> ### Author Response · Authors · 2022-08-02
> **Response to Reviewer 9WbF**
>
> Thank you for the detailed review and your helpful suggestions.
>
> > The implementation of this paper requires several steps, which are somehow distributed in various parts of the paper. A clear algorithm block is missing.
>
> We note that, currently, we have put the Algorithm block in Appendix B. DAE is developed in several steps in the paper and we acknowledge that this can make it difficult to obtain a clear picture of the final algorithm. Provided that the paper gets accepted, we will therefore use the additional page in the camera-ready version to summarize the algorithm and show the algorithm box in the main text.
>
> > Minor: I do not understand the emphasis on some "causal" concepts. These are neither needed (or even used) nor are they any addition to the presented theory.
>
> We emphasize causality because we would like to argue that the advantage is an interesting quantity that arises naturally when one approaches the credit assignment problem from a causal perspective.
> We hope that this connection will inspire future works on more challenging problems (e.g., policy evaluation in POMDPs, where confounders are present), where causal methods are known to be effective.
>
> > L54: Your definition of reward is indeed expected reward (expectation taken oven next state $s'$).
>
> We have addressed this in our revision.
>
> > Paragraph of L150: I generally accept that A is lower variance than Q but using PPO to demonstrate this is not adequate. In particular, I would examine this with a value-based method that directly reinforces Q, instead of AC methods, which normally does not give rise to accurate value function.
>
> We would like to emphasize that these estimates are **not** from the learned value/advantage functions of PPO, but estimated using MC simulations by resetting to the desired state-action pairs and performing rollouts with the policy learned by PPO. While we already explain this experiment in Appendix C.2, we will mention it in the caption of Figure 2 to make the main text more self-contained.
>
> > The term -normalized is somehow misleading, as normalization is a process of mapping to one, not zero. Consider a different term; perhaps something like -neutralized or -nullified.
>
> Thank you for pointing out the misleading naming. We have changed the term to “centered”.
>
> > From proof of theorem 1 (L473), it looks like that non-zero policy is a *missing* assumption, $\pi(s, a)$ for all $(s,a)$ . This kind of assumption is very normal in policy gradient results. I do not consider it as a downside.
>
> We note that in Theorem 1 it is required for $(s, a)$ to be reachable, meaning that both $p(s)$ and $\pi(a|s)$ must be non-zero for $\hat{A}(s, a)$ to converge to $A(s,a)$.
>
> > Using value-function for the N-step case is somehow defeating the purpose as the main premise is to disentangle advantage learning from value learning. Isn't it?
>
> While the bootstrapped method (Theorem 2) also suffers from the same issues mentioned in the motivation, we argue that this can be mitigated by using long-horizon multistep backups and the assumption that actions mostly have short-term effects. For example, consider the idealized case where the influence of an action persists only up to $m$ steps (i.e., the future state distribution is left unchanged by the action after $m$ steps), and a backup horizon of $n > m$. Theorem 2 suggests that the estimated advantage function is truncated after the first $m$ steps, meaning that $V_\text{target}$ would have no effect on the estimated advantage function.
>
> > The last part of the theory (from L232 onwards) is not well-written and requires more explanation and expansion. I recommend removing several parts from the first 4 pages (which are simply reviewing textbook stuff) and expand on the last part of the theory with more scrutiny. In particular, $V_\text{target}$ is not clearly defined and its dependence on learning $V$ is not clear. The trajectory augmentation seems also to be required. From the text, it is very hard to come up with a clear algorithmic machinery. I strongly suggest to add a algorithm block and summarize the details.
>
> Thank you for pointing out that the end of Section 4 could benefit from a more detailed explanation of the algorithm. As already mentioned above, we will use the additional page of the camera-ready revision to expand it and show the algorithm box in the main text. We will cut Section 2 if necessary.
> In response to your questions about the algorithm, we provide a short summary: We integrate DAE into PPO by replacing PPO’s loss function for the value function with DAE’s (Eq 17) and by sampling mini-batches from contiguous parts of trajectories. When updating the policy, we use the learned advantage function $\hat{A}$ instead of GAE. In Theorem 2, $V_\text{target}$ has the same role as in standard TD learning, i.e., it is the bootstrapping target for updating the value function. Details about the practical implementation can be found in Appendix B.

---

> > ### Comment · Reviewer_9WbF · 2022-08-08
> > **Authors' Response**
> >
> > Thanks for the responses. They are helpful [the renaming also makes more sense]. I maintain my positive outlook for acceptance.

---

### Official Review · Reviewer_tFqS · 2022-07-11

**Rating:** 4
**Confidence:** 4
**Soundness:** 1 poor
**Presentation:** 3 good
**Contribution:** 2 fair

**Summary:**

This paper proposes Direct Advantage Estimation, an algorithm which estimates
directly the advantage function instead of the traditional state(-action) value
estimation. They draw some links with a causal interpretation and perform
experiments on a synthetic environment as well as MiniAtari and ALE, comparing
themselves to PPO with GAE.


**Questions:**

- As you are sampling on a trajectory by trajectory basis instead of minibatch of
  frames (l 497): how does your algorithm perform in terms of wallclock time
  compared to PPO?
- Fig 4: Does DAE still have lower variance in a more general setting rather
  than one designed so that DAE can reach 0 variance easily? I am
  thinking a simple gridworld, the Q and V function can be computed in closed form.
- Do you use p for the discounted or undiscounted stationary distribution? It seems in eq 39-41 it would be the discounted if you are using an unbiased estimator of policy gradient, however it seems to be used as the undiscounted in the main text.
- To make sure I understand clearly, what exactly is \hat{A} in alg 1? Is it just the learned advantage or the n-step TD with advantage function?

**Limitations:**

Some limitations are addressed with Fig 3. However I think the authors need to make a stronger case for their method:
- First, a comparison with a more similar algorithm (n-step returns) would be better than GAE, as it would showcase that incorporating the advantage in the TD error is important
- Then they would need to show why estimating directly the advantage as they do is useful/important/interesting. This could be tested experimentally by comparing with other methods for estimating the advantage.

**Strengths And Weaknesses:**

# Strengths
The method proposed is motivated theoretically, is simple to implement and seems
to show improved performance compared to the baseline in the experiments. The
paper is overall easy to read.

# Weaknesses
- The paper displays a collection of different results and interpretations but
  few seems actually necessary to explain the method. More precisely
    1. I find the connection to causality weak overall. In this light any
       centered variable can be thought of as an instance of a causal power, this does not necessarily mean there is any causal estimation behind.
    2. The content of section 3 is mostly motivated by the synthetic example.
       I find this problematic as I believe the synthetic example is a very
       contrived one. While one motivation for DAE is the credit assignment
       problem (l22-23) this environment is specifically designed so there is no
       credit assignment.

- Baselines: while in practice DAE ends up being very close to a n-step return
  method, the only comparison point in the paper is with GAE. A minimum I would
  need to recommend this paper for acceptance would be a fair comparison with
  its 1. n-step return ablation (by removing the effect of A in eq 17 if I
  understood correctly).
  Then I think, to show the interest of directly estimating the advantage,
  another baseline could be to replace your directly learned A by
  2. Q - V where Q and V are learned like usual
  3. A = Q - V learned with a duelling architecture.
  1) would show if your estimator which incoporates the advatange is superior to
     the tradional TD one
    2) and 3) would be useful to show whether learning the advantage directly
       is useful. Indeed while you make a case for the advantage being a quantity of interest, I find there is little evidence in the paper for why learning it directly with your method is important compared to the methods 2) and 3).

Minor points:
- eq 4: maybe rather than the approximate symbol it would be more precise and
  exact to say that it is a sample?
- As you do not seem to be Vtrace or the Impala algorithm, I think using "IMPALA" just to mean the architecture can be misleading.

---

> ### Author Response · Authors · 2022-08-02
> **Response to Reviewer tFqS (1/2)**
>
> Thank you for your feedback and suggestions. We hope we have addressed your concerns with additional experiments, and we are happy to answer additional questions during the discussion phase.
>
> > I find the connection to causality weak overall. In this light any centered variable can be thought of as an instance of a causal power, this does not necessarily mean there is any causal estimation behind.
>
> While the advantage function is simply the Q-function centered by the value function in terms of its definition, we argue in the paper that the advantage function could also arise naturally if one approaches the problem from a causal perspective. Although the connection is not explicitly used in the development of our method, we hope that this connection will inspire future works to approach more difficult problems (e.g. policy evaluation in POMDPs) from a different perspective.
>
> > The content of section 3 is mostly motivated by the synthetic example. I find this problematic as I believe the synthetic example is a very contrived one. While one motivation for DAE is the credit assignment problem (l22-23) this environment is specifically designed so there is no credit assignment.
>
> Although the synthetic example is an extreme case, we do believe that a weaker property, i.e., that the future state distribution depends weakly on the action, is prevalent in more realistic problems. For example, the concept of bottleneck states in hierarchical RL assumes that there are certain states that lead to other densely connected regions (e.g., doorways), and are frequently traversed by the policy. If these bottleneck states are frequently reached, then it suggests that the future state distributions past the bottleneck state have weak dependencies on the actions before the bottleneck state.
>
> > Baselines: while in practice DAE ends up being very close to a n-step return method, the only comparison point in the paper is with GAE. A minimum I would need to recommend this paper for acceptance would be a fair comparison with its 1. n-step return ablation (by removing the effect of A in eq 17 if I understood correctly). Then I think, to show the interest of directly estimating the advantage, another baseline could be to replace your directly learned A by (2) Q - V where Q and V are learned like usual (3) A = Q - V learned with a duelling architecture. (1) would show if your estimator which incoporates the advatange is superior to the tradional TD one. (2) and (3) would be useful to show whether learning the advantage directly is useful. Indeed while you make a case for the advantage being a quantity of interest, I find there is little evidence in the paper for why learning it directly with your method is important compared to the methods 2) and 3).
>
> Thank you for suggesting additional baselines. We are uncertain about the meaning of ablation (1). DAE learns the advantage function by optimizing the loss in Eq 17. By removing $\hat{A}$ from it we are left with no advantage estimates. It is therefore not clear how to estimate the advantage in ablation (1). We would invite you to discuss the idea for ablation (1) with us during the discussion phase.
> We have performed both ablations (2) (denoted “Indirect”) and (3) (denoted “Duel”) in our revision, please see the new section Appendix D for more details.
> Here is a brief summary of our findings:
> 1. Both DAE and Duel perform better than Indirect, suggesting that modeling the advantage function directly can be beneficial.
> 2. DAE performs better than Duel, suggesting that jointly estimating the advantage function across time steps as done in DAE can significantly improve the performance.
>
>
> > As you do not seem to be Vtrace or the Impala algorithm, I think using "IMPALA" just to mean the architecture can be misleading.
>
> We have changed the name from "IMPALA" to "Deep" to avoid confusion.
>
> > As you are sampling on a trajectory by trajectory basis instead of minibatch of frames (l 497): how does your algorithm perform in terms of wallclock time compared to PPO?
>
> The difference to the sampling process in PPO with GAE is that we require the samples to be sequences $(s_t, a_t, r_t, s_{t+1}, a_{t+1}, r_{t+1}, …)$ that are contiguous in time (non-terminated trajectories) to compute the loss. For example, when we say the batch size is 256 and the n-step is 128, we sample in each gradient step 2 sequences of 128-step transitions. Essentially, the only difference in wall-clock time comes from the computation of different loss functions, which we found to be negligible.

---

> > ### Author Response · Authors · 2022-08-02
> > **Response to Reviewer tFqS (2/2)**
> >
> > > Fig 4: Does DAE still have lower variance in a more general setting rather than one designed so that DAE can reach 0 variance easily? I am thinking a simple gridworld, the Q and V function can be computed in closed form.
> >
> > We note that Fig 4 shows the mean squared error between the estimated advantage function and the true advantage function (known in this case). The reason why GAE did not converge to 0 is due to the sampling nature of its estimation. On the other hand, DAE models the advantage function with a deterministic function, and by Theorem 2, it should converge to the true advantage function irrespective of the environment, given enough data and capacity of the approximator. The rate of convergence and its dependency on the environment is an interesting question. However, a detailed analysis of this relationship is beyond the scope of this paper.
> >
> > > Do you use p for the discounted or undiscounted stationary distribution? It seems in eq 39-41 it would be the discounted if you are using an unbiased estimator of policy gradient, however it seems to be used as the undiscounted in the main text.
> >
> > We use $p(s_t = s)$ for the undiscounted probability of being in state $s$ after $t$ time steps. In equation 39 we are interested in the starting state distribution, hence $p(s_0=s)$. We will clarify this at the beginning of the proofs section in the revision. We are uncertain about whether we understand your comment about a connection between equations 39-41 and an estimate of a policy gradient correctly. We would therefore like to invite you to elaborate on this point, and we will try to clarify the remaining questions during the discussion period.
> >
> > > To make sure I understand clearly, what exactly is \hat{A} in alg 1? Is it just the learned advantage or the n-step TD with advantage function?
> >
> > In practice, we use a dueling network to simultaneously approximate the value function and the advantage function, and $\hat{A}$ is simply the output from the advantage stream of the network (i.e., the learned advantage). Appendix C.4.2 contains a discussion of the network architecture (depicted in Figure 6).

---

> > > ### Comment · Reviewer_tFqS · 2022-08-08
> > > **reply**
> > >
> > > Thank you for your answer. While I won't change my score before reviewer/AC discussions I take note of the points you made and of the additional results.
> > >
> > > > While the advantage function is simply the Q-function centered by the value function in terms of its definition, we argue in the paper that the advantage function could also arise naturally if one approaches the problem from a causal perspective. Although the connection is not explicitly used in the development of our method, we hope that this connection will inspire future works to approach more difficult problems (e.g. policy evaluation in POMDPs) from a different perspective.
> > >
> > > Yes but in this work, you are only comparing a value to its expectation (over all the stochasticity) which I think is quite limited for a causality interpretation (where you would usually condition on some realization/event), especially given that there is already work published in that area like Mesnard 2020 which you cited which attempts to learn the counterfactual using a hindsight baseline/advantage.
> > >
> > > > Although the synthetic example is an extreme case, we do believe that a weaker property, i.e., that the future state distribution depends weakly on the action, is prevalent in more realistic problems. For example, the concept of bottleneck states in hierarchical RL assumes that there are certain states that lead to other densely connected regions (e.g., doorways), and are frequently traversed by the policy. If these bottleneck states are frequently reached, then it suggests that the future state distributions past the bottleneck state have weak dependencies on the actions before the bottleneck state.
> > >
> > > Yes exactly, past these bottlenecks, however you could argue that these bottlenecks are at the core of what makes credit assignment difficult. Therefore I am still not convinced by your approach of presenting your work as a causality/credit assignment technique and then choosing examples that ignore this problem.
> > >
> > > > Thank you for suggesting additional baselines. We are uncertain about the meaning of ablation (1). DAE learns the advantage function by optimizing the loss in Eq 17. By removing
> > >  from it we are left with no advantage estimates. It is therefore not clear how to estimate the advantage in ablation (1). We would invite you to discuss the idea for ablation (1) with us during the discussion phase. We have performed both ablations (2) (denoted “Indirect”) and (3) (denoted “Duel”) in our revision, please see the new section Appendix D for more details. Here is a brief summary of our findings:
> > > Both DAE and Duel perform better than Indirect, suggesting that modeling the advantage function directly can be beneficial.
> > > DAE performs better than Duel, suggesting that jointly estimating the advantage function across time steps as done in DAE can significantly improve the performance.
> > >
> > > Yes for (1) I meant simply not using the learned advantage and just using n-step return on the value function $\sum \gamma^t r_t + \gamma^T V(s_T) - V(s_0)$. Thank you for the experiments, the comparison with the duelling advantage is interesting.
> > >
> > > > We note that Fig 4 shows the mean squared error between the estimated advantage function and the true advantage function (known in this case). The reason why GAE did not converge to 0 is due to the sampling nature of its estimation. On the other hand, DAE models the advantage function with a deterministic function
> > >
> > > Ok, what I meant is that as far as I know, all the advantage estimation methods we have suffer from a bias-variance tradeoff. Learning Q and V to estimate A will also work, as DAE, to find an estimator with 0 residual variance. This does not prove however that GAE/n-step returns are inferior in my opinion as depending on the environment they might prove a better estimator of the value/advantages. For instance https://arxiv.org/abs/1806.01175 discusses that and shows that in complex environments longer traces (and therefore which would have more variance in their estimate) can be better.
> > > Therefore I am not convinced that the result in Fig 4 can say anything about the superiority of a method compared to another in a more realistic setting.
> > >
> > > >We use  for the undiscounted probability
> > >
> > > For an unbiased estimator of policy gradient the expectation needs to be on the discounted stationary distribution. Looking back at the proof it's only used in expectation over the value estimation error so it should not change anything I think.
> > > Reading more closely the proof of Thm2, do you think you can go directly from the sum in (34) being 0 to all the terms in (37) being 0? Don't you need an argument that it is true because it holds for all timesteps or something of the sort?

---

> > > > ### Author Response · Authors · 2022-08-09
> > > > **Re:**
> > > >
> > > > Thank you for the constructive feedback.
> > > >
> > > > > Yes but in this work, you are only comparing a value to its expectation (over all the stochasticity) which I think is quite limited for a causality interpretation (where you would usually condition on some realization/event), especially given that there is already work published in that area like Mesnard 2020 which you cited which attempts to learn the counterfactual using a hindsight baseline/advantage.
> > > >
> > > > Despite its simplicity, we believe that the definition of the advantage function does capture a certain aspect of causal effects, as illustrated by the synthetic example. However, we believe that this simplicity is partly because we are focusing on fully observable MDPs, where confounders are not present. We do expect that for more challenging problems, machinery from causal inference (e.g., counterfactuals or interventions) may play a more important role.
> > > >
> > > > > Yes exactly, past these bottlenecks, however you could argue that these bottlenecks are at the core of what makes credit assignment difficult. Therefore I am still not convinced by your approach of presenting your work as a causality/credit assignment technique and then choosing examples that ignore this problem.
> > > >
> > > > While the problem of credit assignment is multifaceted, we do believe that our example captures one interesting aspect of the problem, that is, the ability to discard information that is irrelevant to decision-making. Although we demonstrated this through an extreme example, we also showed empirically that the advantage function is more stable (Fig 2), suggesting that this stability may hold in more realistic settings.
> > > >
> > > > > Yes for (1) I meant simply not using the learned advantage and just using n-step return on the value function . Thank you for the experiments, the comparison with the duelling advantage is interesting.
> > > >
> > > > Thank you for the clarification, however, we note that GAE is equivalent to n-step method when $\lambda=1$, and the effect of $\lambda$ was studied in the GAE paper [1] where they found that $\lambda\in[0.9, 0.99]$ is usually optimal. In another large-scale study [2], the authors also found that GAE (with $\lambda<1$) outperforms n-step methods most of the time and is recommended as the default. We, therefore, did not include an n-step baseline, as it can be expected to be considerably weaker than GAE with a tuned $\lambda$.
> > > >
> > > > [1] Schulman, J., Moritz, P., Levine, S., Jordan, M., & Abbeel, P. (2015).
> > > > High-dimensional continuous control using generalized advantage estimation. ArXiv:1506.02438. https://arxiv.org/abs/1506.02438
> > > >
> > > > [2] Andrychowicz, M., Raichuk, A., Stańczyk, P., Orsini, M., Girgin, S., Marinier, R., Hussenot, L., Geist, M., Pietquin, O., Michalski, M., Gelly, S., & Bachem, O. (2020). What Matters In On-Policy Reinforcement Learning? A Large-Scale Empirical Study. ArXiv:2006.05990. http://arxiv.org/abs/2006.05990
> > > >
> > > > > Ok, what I meant is that as far as I know, all the advantage estimation methods we have suffer from a bias-variance tradeoff. Learning Q and V to estimate A will also work, as DAE, to find an estimator with 0 residual variance. This does not prove however that GAE/n-step returns are inferior in my opinion as depending on the environment they might prove a better estimator of the value/advantages. For instance https://arxiv.org/abs/1806.01175 discusses that and shows that in complex environments longer traces (and therefore which would have more variance in their estimate) can be better. Therefore I am not convinced that the result in Fig 4 can say anything about the superiority of a method compared to another in a more realistic setting.
> > > >
> > > > The bias-variance tradeoff certainly applies to DAE as well. Consider the example when the function approximator is trivial (i.e., A(s,a) = 0, a function that always outputs zero), then the DAE estimates would be high bias but low (zero) variance, while GAE, on the other hand, can still perform meaningful estimations due to the use of unbiased n-step estimators. A similar phenomenon is also observed in Fig 4, where at the beginning of training, DAE is heavily biased and underperforms GAE in terms of policy optimization. We do not claim that Fig. 4 directly implies superior performance in more realistic settings, and consequently evaluate DAE empirically on MinAtar and ALE in the Experiments section.
> > > >
> > > > > Reading more closely the proof of Thm2, do you think you can go directly from the sum in (34) being 0 to all the terms in (37) being 0? Don't you need an argument that it is true because it holds for all timesteps or something of the sort?
> > > >
> > > > We note that the sum (Eq 30 in the latest revision) is equal to
> > > > $\sum_{k=0}^t\gamma^k p(s_k{=}s')\left(\gamma^k (Q(s’,a’)-f(s’,a’)-V(s’)\right) = 0$, where the term $Q(s’,a’) - f(s’, a’) - V(s’)$ is independent of $k$. Therefore, if $p(s_k=s’)>0$ for some $k$ (i.e., reachable), then $Q(s’,a’) - f(s’,a’) - V(s’)$ must be 0.

---

### Official Review · Reviewer_8k7E · 2022-07-11

**Rating:** 7
**Confidence:** 4
**Soundness:** 4 excellent
**Presentation:** 4 excellent
**Contribution:** 3 good

**Summary:**

This paper outlines a connection between the causality literature and the advantage function in reinforcement learning. The advantage function can be viewed as an estimate of the causal effect of an action on the expected return. As a second contribution, a theorem shows that the advantage function can be viewed as the solution of a variance-minimization problem subject to a certain constraint. This leads to a different objective for learning the advantage function without resorting to estimating the Q and V value functions. This approach is shown to outperform generalized advantage estimation (GAE) in a suite of standard benchmark tasks.

**Questions:**

- Table 1 presents counts for how often an algorithm was better (DAE or GAE). Does this account for the confidence intervals? As a rule-of-thumb, if the confidence intervals overlap, it wouldn't be fair to say one was better than the other. This table could include a number for the the number of environments with approximately equal performance.

- The paper makes a connection between the advantage function and causal effects but, afterwards, doesn't make use of this link. Do you have any other thoughts on how this interpretation could lead to better RL algorithms?

- Concerning Fig. 2, do you have an explanation why the advantage function tends be more stable? Do you think it's related to Prop. 1 and actions have limited effects across time in these RL environments?
Also, have you looked at the plot of Fig. 2 for other environments? Do you still find the same thing?

- Lines 162-166: Is it true that in a multi-step setting, the advantage function would be more stable? It seems like delayed rewards are an important factor in explaining the difference between Q-values and advantage functions.

- Any thoughts on viewing the advantage-function as a variance-minimizing state-action-dependent baseline (with a constraint)? How would you compare that to the solution of the unconstrained version of the same optimization problem?



**Limitations:**

Yes, I don't see any major issues here.

**Strengths And Weaknesses:**

The paper has two distinct contributions: the connection of advantage functions to causal effects and the new objective for estimating the advantage function. I find both of these to be interesting and of interest to the community.
Moreover, the paper is well-written and easy to follow, from the organization to the choice of notation.

The exposition of the benefits of the advantage function over Q-values was well-done. The intuitions was nicely explained and supported by the synthetic experiment and Fig. 2.
After checking some proofs, they seem to be correct and are adequately explained. I had no trouble following them. To me, it's a nice result that you can view the advantage function as a variance-minimizing state-action-dependent baseline (under some constraint).
The experiments seem sound with reasonable choices as baselines and good experimental procedures. Sufficient details are presented in the appendix to understand the experiements in-depth. In particular, I thought the shuffling of actions in the synthetic experiment (as described in the appendix) was nice experimental design. The experiment varying the size of the network  also yielded some interesting results.

Although I am not very familiar with the causality literature, the link between causal effects and advantage functions is intriguing to me. While this connection wasn't developed too thoroughly in this work, it does outline a direction that could be explored further in the future.
Overall, I think this is a solid paper and haven't found any major issues, although I do have some questions which I've written in the next section.

---

> ### Author Response · Authors · 2022-08-02
> **Response to Reviewer 8k7E**
>
> Thank you for your thorough review and your constructive feedback.
>
> > Table 1 presents counts for how often an algorithm was better (DAE or GAE). Does this account for the confidence intervals? As a rule-of-thumb, if the confidence intervals overlap, it wouldn't be fair to say one was better than the other. This table could include a number for the number of environments with approximately equal performance.
>
> Based on your suggestion, we have added a “similar” row in Table 1 to account for scores that are within 1 standard error of the mean, and we note that DAE still outperforms GAE in most environments.
>
> > The paper makes a connection between the advantage function and causal effects but, afterwards, doesn't make use of this link. Do you have any other thoughts on how this interpretation could lead to better RL algorithms?
>
> While we use the link between the advantage function and causal effects mostly to motivate Direct Advantage Estimation, we believe that the advantage function deserves more attention and that there is more to it than “the Q-function minus the baseline value function”. One interesting direction related to the causal effects interpretation is policy evaluation in confounded environments (e.g., POMDP), as causal methods are known to be effective when confounders are present. An analysis of this problem is beyond the scope of this work, however.
>
> > Concerning Fig. 2, do you have an explanation why the advantage function tends be more stable? Do you think it's related to Prop. 1 and actions have limited effects across time in these RL environments? Also, have you looked at the plot of Fig. 2 for other environments? Do you still find the same thing?
>
> Yes, we believe that the advantage function tends to be more stable because actions have short-term effects across time. We’ve included another plot for the Asterix environment from MinAtar in the appendix and, again, we see a similar pattern, i.e., the advantage function is more stable than the Q-function. See Fig. 9 in Appendix E  for the result.
>
> > Lines 162-166: Is it true that in a multi-step setting, the advantage function would be more stable? It seems like delayed rewards are an important factor in explaining the difference between Q-values and advantage functions.
>
> That’s correct, the major difference is in how distant rewards are attributed. We would say that the advantage function is more stable when actions in the given environment tend to have short-term impacts on the future state distributions. The motivating example is an extreme case where actions have no effect on future state distributions. A more realistic example would be environments with bottleneck states (states that connect other densely connected regions of states). In this case, if the bottleneck states are typically reached with weak dependencies on the actions chosen, then the advantage function would also be stable. On the other hand, in a tree-like environment where actions lead to completely disconnected branches, the advantage function would also suffer from the same problem as the Q-function.
>
> > Any thoughts on viewing the advantage-function as a variance-minimizing state-action-dependent baseline (with a constraint)? How would you compare that to the solution of the unconstrained version of the same optimization problem?
>
> In the unconstrained case, the problem would become underdetermined. Consider the setting of Theorem 2, one can set $\hat{A}(s,a)$ to $r(s,a)$ and $V(s) = V_\text{target} = 0$ to achieve zero loss. More generally, any function $\hat{A}(s,a) = r(s,a) + E[\phi(s’) | s, a] - \phi(s)$ and $V(s) = \phi(s)$, where $\phi(s)$ is an arbitrary function of the state would lead to zero loss. An intuitive explanation for why the constraint is necessary is that when enforcing it, $\phi$ has to satisfy the equation $\sum_a \pi(a|s) \hat{A}(s,a) = E[r(s,a) + \phi(s’)|s] - \phi(s) = 0$, meaning that $\phi$ is the unique solution to the Bellman equation, which is the value function.

---

> > ### Comment · Reviewer_8k7E · 2022-08-09
> > **Reply**
> >
> > Thank you for the clarifications and additions to the paper. Your response about understanding the need for the constraint was helpful.
> > I'm happy to still recommend acceptance.

---

### Author Response · Authors · 2022-08-02
**Response to all reviewers**

We thank the reviewers for their insightful comments and helpful suggestions, which greatly helped us improve our paper. We appreciate that our work was perceived as “novel and useful” (Reviewer 9WbF), “fundamental yet accessible” (Reviewer c2up), and “of interest to the community” (Reviewer 8k7E).

Based on the suggestions from Reviewer tFqS, we have included an additional ablation study in Appendix D to demonstrate the effectiveness of DAE.
We have also responded to the questions and suggestions individually, and are looking forward to a constructive discussion phase.

---

### Meta-Review · Area_Chair_Q1f3 · 2022-08-27

**Recommendation:** Accept
**Confidence:** Certain

**Metareview:**

Overall, the proposed approach is well-motivated, simple to implement and appears to work well empirically. The connection between the advantage function and causal concepts is appreciated by some of the reviewers, but a point of confusion and viewed as insufficiently motivated by others. I recommend acceptance for this paper, but for the causal connection, I want to ask the authors to either provide more in-dept explanation or to de-emphasize this connection.

**Award:**

No

---

### Decision · Program_Chairs · 2022-09-14

Accept